

# On the Mechanisms Driving Latent Heat Flux Variations in the Northwest Tropical Atlantic: a Modeling Approach

Pablo Fernández[1], Sabrina Speich[2], Carlos Conejero[3,4], Lionel Renault[3], Fabien Desbiolles[3], Claudia Pasquero[5], and Guillaume Lapeyre[2]

[1]Laboratoire d'Océanographie et du Climat: Expérimentations et Approches Numériques, Sorbonne Université, Paris, France.
[2]Laboratoire de Météorologie Dynamique, École Normale Supérieure, Department of Geosciences, Paris, France.
[3]LEGOS, University of Toulouse, IRD, CNRS, CNES, UPS, Toulouse, France
[4]ENTROPIE (IRD, CNRS, Ifremer, Université de la Réunion, Université de la Nouvelle-Calédonie), Nouméa, New Caledonia.
[5]Department of Earth and Environmental Sciences, University of Milano - Bicocca, Milan, Italy.

**Correspondence:** Pablo Fernández (pablo.fernandez-fernandez@locean.ipsl.fr)

**Abstract.** In this study, a high-resolution ocean-atmosphere coupled simulation is used to assess the effects of sea surface temperature (SST), surface currents, and ocean vertical stratification on the spatial variability of latent heat flux (LHF) and the stability of the marine atmospheric boundary layer (MABL) in the Northwest Tropical Atlantic during January and February 2020. The analysis focuses on the ocean mesoscale ($O(50-250$ km$)$) across the Northwest Tropical Atlantic (referred to as
the EURECA region in this study) and within three sub-regions characterized by different ocean dynamical regimes: Amazon, Downstream, and Tradewind. Results indicate that the coupling between SST and wind speed (and specific humidity) is stronger (weaker) in the Amazon and Downstream regions (influenced by the warm coastal North Brazil Current eddy corridor and the Amazon river plume) than in the Tradewind region (representative of the open ocean), consistent with previous remote sensing studies. Overall, warmer SSTs are associated with increased wind speeds and variations in specific humidity, deviating
from Clausius-Clapeyron expectations. We interpret this as the result of active ocean processes modifying the near-surface atmosphere, enhancing vertical motion in the MABL, and transporting momentum and drier air from the free troposphere toward the surface. This effect is particularly pronounced over waters influenced by the Amazon plume, where positive SST anomalies persist, primarily due to lateral advection in the mixed layer. To further investigate the impact of mesoscale SST features on LHF, we apply a linear, SST-based downscaling method. Results show that these mesoscale SST structures induce
a substantial increase in LHF, 46.8 W m$^{-2}$ K$^{-1}$ on average in the Amazon and Downstream regions (warm eddy corridor). In the Tradewind region, the LHF sensitivity to SST is smaller, at about 35 W m$^{-2}$ K$^{-1}$. For the Amazon region, of the 46.7 W m$^{-2}$ K$^{-1}$ change in LHF associated with SST, approximately 7.8 W m$^{-2}$ K$^{-1}$ is attributed to direct mesoscale SST changes (thermodynamic contribution), while the remainder is linked to mesoscale SST-induced modifications in near-surface atmospheric circulation (dynamic contribution). Within the dynamic contribution, about 80% (31.1 W m$^{-2}$ K$^{-1}$ out
of 38.9 W m$^{-2}$ K$^{-1}$) is due to variations in specific humidity undersaturation, and the remaining 20% (7.8 W m$^{-2}$ K$^{-1}$ out of 38.9 W m$^{-2}$ K$^{-1}$) is due to wind speed changes. Similar relative contributions are found in the other subregions and in the overall EURECA domain. Finally, the influence of surface currents on winds is weaker, with LHF deviations not exceeding



15 W m$^{-2}$. This study underscores the importance of a regionalized approach to mesoscale air-sea interaction studies in the Northwest Tropical Atlantic, as LHF sensitivity to SST and surface currents exhibits strong spatial variability driven by distinct oceanic dynamics. Submesoscale LHF sensitivity to SST and currents is not addressed here and will be the subject of future research.

# 1 Introduction

Turbulent heat fluxes (THFs) are related to temperature (sensible) and moisture undersaturation (latent) imbalances at the air-sea interface. When examining air-sea interactions through THFs, it is common in the literature to distinguish between the ocean's large-scale and *fine-scale* processes, which include the mesoscale ($O(50-250$ km$))$ and submesoscale ($< O(50$ km$))$ components. These *fine-scale* interactions with the atmosphere have been shown to differ significantly from large-scale processes (Chelton and Xie, 2010; Small et al., 2019; Gentemann et al., 2020; Conejero et al., 2024). At large scales, atmospheric dynamics predominantly drive ocean variability (Gill and Adrian, 1982). However, at scales smaller than 250 km, the ocean actively influences the near-surface atmosphere, affecting air temperature, frictional stress, and the stability of the marine atmospheric boundary layer (MABL) (Small et al., 2008). Among THFs, this study focuses on latent heat flux (LHF) in the Northwest Tropical Atlantic, as it provides a direct link between atmospheric dynamics and thermodynamics.

The effects of *fine-scale* sea-surface temperature (SST) variability on the near-surface atmosphere and air-sea heat fluxes, particularly LHF, have been investigated using satellite products (Bishop et al., 2017; Fernández et al., 2023), *in-situ* measurements (Acquistapace et al., 2022; Iyer et al., 2022; Fernández et al., 2024), and atmospheric models (Borgnino et al., 2025). In the literature, two primary mechanisms of lower atmospheric response to SST features have been identified: the downward momentum mixing (DMM) (Hayes et al., 1989; Wallace et al., 1989) and the pressure adjustment (PA) (Lindzen and Nigam, 1987). In the DMM mechanism (Fig. 1a), a warm SST anomaly destabilizes the MABL, enhancing vertical mixing. This process facilitates the entrainment of drier air from the free troposphere into the MABL, thereby increasing surface winds and intensifying LHF (Acquistapace et al., 2022; Borgnino et al., 2025). Conversely, cold SST anomalies suppress vertical mixing, leading to reduced surface winds and lower LHF. Thus, DMM provides a *top-down* mechanism by which *fine-scale* SST variability influences the near-surface atmosphere, a process known as thermal feedback (TFB) (Renault et al., 2019b, 2023). PA, on the other hand, predicts that surface wind convergence (divergence) occurs over SST maxima (minima) as warm (cold) SST cores generate local sea level pressure lows (highs). This leads to weaker winds over SST extrema, resulting in lower LHF (Pasquero et al., 2021). The influence of these mechanisms has been observed across various regions of the World Ocean on timescales ranging from hours to weeks. Foussard et al. (2019) found that PA tends to dominate where surface winds are well-coupled to upper-level winds, while DMM prevails in neutrally stable lower tropospheric conditions where SST effectively modulates surface winds (Desbiolles et al., 2023). Warm mesoscale eddies have been shown to enhance LHF via DMM in several regions, including the Gulf Stream (Minobe et al., 2008), the Kuroshio Extension (Xu et al., 2011; Ma et al., 2015; Chen et al., 2017), the South China Sea (Liu et al., 2018, 2020), and the Agulhas (O'Neill et al., 2005) and Malvinas (Villas Bôas et al., 2015; Leyba et al., 2017) currents. Meanwhile, PA has been shown to impact cloud and precipitation patterns



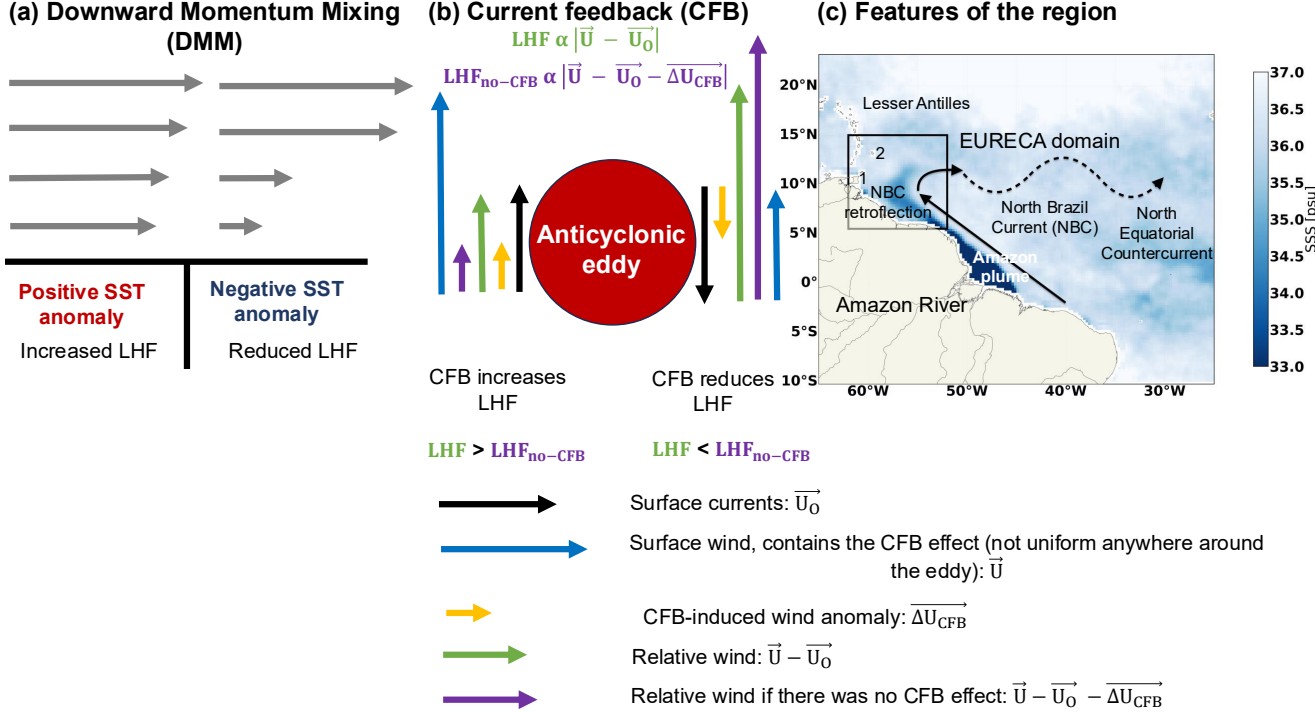

**Figure 1.** (a) Schematic representation of the Downward Momentum Mixing (DMM) mechanism, adapted from Meroni et al. (2020). (b) Schematic representation of the Current Feedback (CFB) mechanism. Details on how to compute the CFB-induced wind anomaly ($\Delta U_{CFB}$) are provided in subsection 3.1. (c) Major dynamical features of the western equatorial Atlantic (arrows) overlaid on the averaged sea-surface salinity (SSS) field from February $17^{th}$–$19^{th}$, 2020. The SSS field is derived from the SMAP-SSS Level 3, version 4.0, 8-day running-mean gridded product (Boutin et al., 2021). The black box delineates the EURECA region, which corresponds to the simulation domain used in this study. Numbers 1 and 2 mark the locations of Trinidad and Tobago and a region near Barbados, respectively. These geographical references, along with the Lesser Antilles, are used throughout the main text.

in the cold wake of tropical cyclones (Ma et al., 2020) through a cross-track secondary circulation (Pasquero et al., 2021). Finally, Conejero et al. (2025) showed that both PA and DMM are diminished when the model spatial resolution is increased and the full range of mesoscale structures is resolved due to a stronger submesoscale-induced atmospheric frontogenesis. In the Northwest Tropical Atlantic, Fernández et al. (2023) found that DMM dominates over PA using multiple satellite products.

In addition to SST effects, surface currents, also influence wind stress and thus surface winds and LHF via a *bottom-up* process known as current feedback (CFB), (Bye, 1985; Chelton et al., 2001; Renault et al., 2016b, a, 2019a). Here, we focus on the CFB-induced surface wind response and its impact on LHF. When surface currents and winds are aligned, an anomalous surface wind develops, reinforcing the prevailing wind field (Renault et al., 2016b). This enhances relative wind speed (the difference between surface wind and surface current velocities) and increases LHF. Conversely, when surface currents oppose



surface winds, the CFB-induced wind anomaly weakens surface winds, which could lead to smaller LHF. This process is illustrated in the left and right sides of the eddy in Fig. 1b respectively.

The significance of CFB in eddy dynamics has been highlighted in studies using remote sensing data (Gaube et al., 2015) and high-resolution coupled simulations (Eden and Dietze, 2009; Renault et al., 2016b). Ignoring atmospheric adjustments to CFB might lead to an overestimation of eddy attenuation timescales and an underestimation of eddy amplitude and azimuthal

speed. CFB might also shortens eddy lifetimes, as the observed composite life cycle reconstructed from satellite altimetry in (Renault et al., 2016b) and consisting of a rapid early intensification, a prolonged slow CFB-induced decay, and an abrupt collapse at the end seems to suggest. This would imply that CFB systematically acts as an *eddy-killer*, transferring energy from mesoscale eddies to the atmosphere (Anderson et al., 2011; Bye, 1985; Dewar and Flierl, 1987; Oerder et al., 2018; Renault et al., 2016b). Furthermore, an inaccurate representation of CFB in numerical models has implications for biogeochemistry. In

oligotrophic regions, mesoscale processes enhance the upward transport of limiting nutrients, supporting biological production (Martin and Richards, 2001; Gaube et al., 2013). In coastal upwelling systems, such as along the Californian coast, eddies modulate biological productivity by subducting nutrients below the euphotic zone and advecting biogeochemical material offshore (Gruber et al., 2011; Nagai et al., 2015; Renault et al., 2016a). Misrepresenting CFB may lead to an overestimation of nutrient quenching and offshore transport, thereby impacting marine ecosystems by altering eddy amplitude, lifetime, and

spatial extent.

As stated above, this paper is focused in the Northwest Tropical Atlantic. The ocean circulation in this region, particularly near the Amazon River estuary, is dominated by the North Brazil Current (NBC), as illustrated in Fig. 1c. The NBC is a strong western boundary current originating in the Equatorial and South Atlantic. Around 8°N, it separates from the coast, forming the NBC retroflection, which feeds the North Equatorial Countercurrent. The NBC system is closely linked to two major processes:

the Amazon freshwater advection in the open-ocean (Reverdin et al., 2021) and the periodic formation of mesoscale eddies known as NBC rings (Johns et al., 1990; Richardson et al., 1994). These processes are interconnected, as NBC rings facilitate the offshore transport and the lateral spreading of the Amazon river plume induced by mesoscale advection and smaller-scale instabilities (Johns et al., 1990; Fratantoni and Glickson, 2002; Reverdin et al., 2021; Olivier et al., 2022; Coadou-Chaventon et al., 2024). This results in strong spatial heterogeneity in sea surface salinity (SSS, Fig. 1c), which also influences upper

ocean temperature by modulating stratification. When low salinity dominates the upper-ocean stratification, the ocean mixed layer (ML), the surface layer in direct contact with the atmosphere, becomes shallower. This can lead to the formation of barrier layers (BLs), which promote temperature inversions (Anderson et al., 1996; de Boyer Montégut et al., 2007; Foltz and McPhaden, 2009; Vialard and Delecluse, 1998; Mignot et al., 2012; Mahadevan et al., 2016; Krishnamohan et al., 2019; Coadou-Chaventon et al., 2024). In the presence of a BL, heat and momentum fluxes remain confined to the shallow ML,

which responds more rapidly to atmospheric forcing. Consequently, the ML cools more quickly in winter and warms more rapidly in summer due to the inhibited exchange with deeper ocean layers (Miller, 1976; Sprintall and Tomczak, 1992). This leads to negative (positive) sea surface temperature (SST) anomalies relative to surrounding waters over the Amazon plume and, in turn, reduced (increased) LHF. However, the magnitude of this response remains debated. While observational studies



suggest that BLs have a strong impact on SST (Pailler et al., 1999; Foltz and McPhaden, 2009), numerical models often fail to
reproduce this effect (Breugem et al., 2008; Balaguru et al., 2012; Hernandez et al., 2016).

The objective of this study is to investigate how these processes (TFB and CFB) together with the Amazon plume affect
LHF variations in the Northwest Tropical Atlantic, complementing previous research based on satellite/reanalysis products
(Fernández et al., 2023) and *in-situ* observations (Fernández et al., 2024). To achieve this, we utilize the high-resolution regional
coupled simulation described in Conejero et al. (2024). This research is conducted within the framework of the *Elucidating
the Role of Cloud-Circulation Coupling in Climate – Ocean Atmosphere* (EUREC[4]A-OA) (www.eurec4a.eu) and the *Atlantic
Tradewind Ocean-Atmosphere Mesoscale Interaction Campaign* (ATOMIC) (https://psl.noaa.gov/atomic/) field experiments.
The paper is structured as follows: Section 2 describes the simulation configuration. Section 3 presents the methods used to
analyze model data. The main results are discussed in Section 4, followed by conclusions and future perspectives in Section 5.

## 2 Data

This study utilizes the EURECA ocean-atmosphere coupled simulation (Conejero et al., 2024). The ocean component is based
on the Coastal and Regional Ocean COmmunity (CROCO) model (Shchepetkin and McWilliams, 2005; Debreu et al., 2012),
while the atmospheric component employs the Weather Research and Forecasting (WRF) model (Skamarock et al., 2008).
The two models are coupled via OASIS (Craig et al., 2017), which performs the grid interpolation and temporal averaging for
property exchanges between the two model components every hour. Specifically, the ocean model provides SST and surface
currents to the atmosphere, while WRF returns surface heat, momentum, and water fluxes to CROCO. Further details on the
EURECA simulation configuration are available in Conejero et al. (2024).

The EURECA simulation spans from January 2019 to June 2020, though this study focuses on the January–February 2020
(JF) period. The CROCO domain extends from $5.5°$N to $15.5°$N and $62°$W to $52°$W, with a horizontal resolution of 1 km. Initial
and lateral open boundary conditions are provided by the "Antilles" simulation, which employs the same coupled configuration
but at a coarser resolution over a larger domain, including parts of the Caribbean Sea (Conejero et al., 2025).

The WRF domain in EURECA is slightly larger than the ocean domain to mitigate sponge effects. The atmospheric model
runs at a horizontal resolution of approximately 2 km, with outputs available on 40 $\eta$ vertical levels. To analyze the first 2000 m
of the atmosphere, these levels are linearly interpolated to uniform 100 m vertical spacing. Initial and boundary conditions,
provided by the "Antilles" simulation, are updated every 3 hours. Bulk formulations (Fairall et al., 2003) are used to estimate
freshwater, turbulent, and momentum fluxes, which are then fed into the ocean model. The CFB effect is implemented in both
the surface and planetary boundary layer schemes, following Renault et al. (2019a). WRF variables at multiple vertical levels
are stored every 3 hours, while surface variables are recorded hourly.





## 3 Methodology

Following Fernández et al. (2024), we remove the diurnal cycle from all variables before analyzing LHF sensitivity to the
surface ocean. This is achieved by computing daily means from the WRF/CROCO outputs, and all subsequent calculations are
performed on these averaged variables. It is important to note that this procedure filters out part of the ocean submesoscale
variability. Consequently, this paper deals exclusively with the ocean mesoscale. We briefly return to this point in Section 5.

All analyses are performed in four regions. First, we consider the full simulation domain, referred to as the EURECA domain
$(5.5°-15°N, 62°-52°W)$. Additionally, we examine three distinct sub-regions: Amazon $(7°-9°N, 56°-53°W)$, Downstream
$(9°-11°N, 59°-56°W)$ and Tradewind $(12°-14°N, 58°-52.5°W)$. Fig. 2 displays three black boxes delineating the sub-
regions. The three subdomains are characterized by different ocean dynamics and air-sea interactions. The Tradewind sub-
region, representative of the open ocean, is relatively quiescent, whereas the Downstream sub-region, closer to the coast,
exhibits enhanced *fine-scale* ocean activity, and the Amazon sub-region is influenced by the Amazon plume. These regional
differences have been highlighted in previous studies based on remote sensing and reanalysis data (Fernández et al., 2023) as
well as *in-situ* observations (Fernández et al., 2024). To ensure consistency with these studies, all analyses are restricted to the
January–February (JF) 2020 period.

### 3.1 Coupling Coefficients

Following Renault et al. (2016b, 2019b) and Conejero et al. (2024), we estimate several air-sea coupling coefficients as the
statistically significant slope (determined via a two-sided *t*-test) of the linear regression between the binned distributions of
mesoscale anomalies from two variables. For all coefficient calculations, we exclude mesoscale anomalies located on the
continental shelf (regions where the seafloor is shallower than 100 m) and the Lesser Antilles (west of $60.25°W$), as these
areas can be affected by orography, land-sea interactions, and coastline effects (Desbiolles et al., 2014).

To compute mesoscale anomalies we use a combination of time and spatial filters. In order to remove weather-induced
synoptic variability from the atmospheric variables (winds, specific humidity, and air temperature), we first apply a 29-day
running mean as in Chelton et al. (2007); Renault et al. (2019b); Conejero et al. (2024). To isolate the mesoscale band, we
apply a band-pass isotropic Gaussian filter with a 50–250 km cutoff length (Renault et al., 2019b), to the 29-day running mean
dataset and keep the scales between 50 and 250 km. After performing the filtering for the whole EURECA domain, we select
the anomalies in Amazon, Downstream and Tradewind to operate on them separately.

To characterize the effect of surface currents on near-surface winds (CFB) we compute $s_w$, the coupling coefficient between
surface current vorticity and surface wind curl anomalies (Renault et al., 2016b, 2019b). Note that, since mesoscale currents
are nearly in geostrophic balance (and therefore non-divergent), $s_w$ effectively isolates the CFB from the TFB at the mesoscale
(Renault et al., 2019b). Using $s_w$, the CFB-induced wind anomaly $(\overrightarrow{\Delta U_{CFB}})$ reads:

$$\overrightarrow{\Delta U_{CFB}} = s_w \overrightarrow{U_o}, \tag{1}$$

where $\overrightarrow{U_o}$ stands for surface currents. Recall that $\overrightarrow{\Delta U_{CFB}}$ is represented in orange arrows in Fig. 1b.



| Coupling Coefficient | Description |
|:---:|:---:|
| $s_w$ | surface current vorticity and surface wind curl |
| $s_u$ | SST and surface wind magnitude |
| $s_q$ | SST and surface specific humidity |
| $s_t$ | SST and surface temperature |

**Table 1.** Overview of the coupling coefficients.

Finally, we also calculate the coupling coefficients of near-surface wind speed ($s_u$), near-surface specific humidity ($s_q$) and near-surface atmospheric temperature ($s_t$) mesoscale anomalies with respect to SST mesoscale anomalies. Table 1 provides a summary of the coupling coefficients described above.

## 3.2   LHF Sensitivity to SST and Surface Currents

To evaluate how LHF responds to SST and surface currents, we compute multiple LHF datasets using the COARE3.5 algorithm

(Edson et al., 2013). First, we calculate $\mathrm{LHF_U}$, which represents the LHF dataset obtained using the atmospheric variables of the first WRF vertical level (surface winds, air temperature and specific humidity) together with SST. Note that we do not consider relative winds (i.e. the difference between surface winds and surface currents) to compute $\mathrm{LHF_U}$. Additionally, we compute $\mathrm{LHF_{LR}}$, which corresponds to LHF computed with the *smoothed* same variables as $\mathrm{LHF_U}$ , obtained by applying a Gaussian low-pass filter (cutoff length of 250 km) to the original WRF variables and SST. Again, we do not consider relative

winds in the calculation of $\mathrm{LHF_{LR}}$ either.

We then apply the LHF downscaling algorithm developed by Fernández et al. (2023). Given a *smoothed* variable ($\Psi_{\mathrm{LR}}$), we reconstruct a new dataset incorporating the finer-scale SST features ($\Psi_{\mathrm{HR}}$) as:

$$\Psi_{\mathrm{HR}} = \Psi_{\mathrm{LR}} + s_\psi \Delta \mathrm{SST}. \tag{2}$$

Here, $\Delta \mathrm{SST}$ represents the SST correction, which accounts for deviations of the high-resolution SST field from the coarse

*smoothed* SSTs. To ensure that the domain-averaged SST correction is zero and that the area-weighted means of the variables remain conserved, $\Delta \mathrm{SST}$ is computed as:

$$\Delta \mathrm{SST} = \left( \mathrm{SST} - \overline{\mathrm{SST}} \right) - \left( \mathrm{SST_{LR}} - \overline{\mathrm{SST_{LR}}} \right). \tag{3}$$

$\overline{\mathrm{Overbars}}$ denote the spatial average over the region of study. Thus, to compute $\mathrm{LHF_{HR}}$, we statistically downscale each of the variables driving LHF (surface wind, specific humidity air temperature and SST) using its corresponding coupling

coefficient, as detailed above, and we then apply COARE3.5 with the downscaled variables. Note that we do not use relative winds for $\mathrm{LHF_{HR}}$ either.



To isolate the thermodynamic contribution (i.e. LHF variations solely due to SST changes via modifications of the saturation specific humidity) to LHF sensitivity, we compute an additional LHF dataset, denoted as $\mathrm{LHF_{therm}}$. In this dataset, air temperature and SST are downscaled, while wind speed remains with its *smoothed* value. The specific humidity required to obtain
$\mathrm{LHF_{therm}}$ is derived from the *smoothed* relative humidity and the specific humidity of saturation computed with $\mathrm{SST_{HR}}$ via the Clausius-Clapeyron equation. This ensures that the specific humidity variations for this LHF subset are only due to SST mesoscale changes themselves without the effects of the atmospheric-induced modifications of mesoscale SST structures (i.e. entrainment of drier air from the free troposphere associated with DMM).

Moreover, we compute $\mathrm{LHF_{therm-U}}$ to further distinguish, within the dynamic contribution, the effects of wind speed and
specific humidity. Recall that the dynamic contribution represents LHF changes associated with the mesoscale SST-induced modifications of the near-surface atmosphere (winds and specific humidity). Thus, in $\mathrm{LHF_{therm-U}}$, air temperature, and surface winds are downscaled, while specific humidity is obtained as in $\mathrm{LHF_{therm}}$. This means that the only change between $\mathrm{LHF_{HR}}$ and $\mathrm{LHF_{therm-U}}$ is that specific humidity in the latter is scaled maintaining relative humidity at its *smoothed* value whereas in the former, specific humidity is downscaled with Eq. 2. Thus, $\mathrm{LHF_{therm-U}}$ accounts for both the thermodynamic contribution
to LHF sensitivity to SST and the effect of SST-induced mesoscale surface wind changes.

Finally, to assess the contributions of surface currents in LHF variations, we consider two additional LHF datasets. $\mathrm{LHF_{orig}}$ is computed using the original WRF variables in COARE3.5, but replacing the first vertical model-level wind speed with relative winds. Thus, the difference $\mathrm{LHF_{orig} - LHF_{U}}$ represents the variations in LHF associated with the consideration of relative winds instead of just surface winds when computing LHF. Meanwhile, $\mathrm{LHF_{no-CFB}}$ stands for LHF computed with
relative winds, but with the CFB-induced wind anomaly ($\overrightarrow{\mathrm{\Delta U_{CFB}}}$, Eq. 1, represented in orange arrows in Fig. 1b) removed component-wise. Hence, $\mathrm{LHF_{orig} - LHF_{no-CFB}}$ contains the CFB effect on LHF.

### 3.3 Mixed Layer Heat Budget

To provide a broader insight on the linkages between the mesoscale SST anomalies leading to LHF variations and the Amazon River plume, we present a mixed layer heat budget analysis in the Amazon sub-region. The temperature equation, vertically
integrated down to the mixed layer depth reads (Vialard and Delecluse, 1998):

$$\underbrace{\langle\partial_t\mathrm{T}\rangle_\mathrm{H}}_{\text{Total tendency}} = \underbrace{\langle-\mathrm{u}\partial_x\mathrm{T} - \mathrm{v}\partial_y\mathrm{T}\rangle_\mathrm{H}}_{\text{Horizontal advection}} - \underbrace{\langle\mathrm{w}\partial_z\mathrm{T}\rangle_\mathrm{H}}_{\text{Vertical advection}} + \underbrace{\overbrace{\frac{\mathrm{Q_s}\left(1-\mathrm{F_H}\right)+\mathrm{Q_{ns}}}{\rho_0\mathrm{C_pH}}}^{\text{Forcing }\mathcal{F}}}_{\text{Atmospheric forcing}} +$$
$$+ \underbrace{\overbrace{\frac{\partial_t\mathrm{H}}{\mathrm{H}}\left(\mathrm{T_H} - \langle\mathrm{T}\rangle_\mathrm{H}\right)}^{\text{Mixing }\mathcal{D}}}_{\text{Entrainment}} + \text{Residual} \qquad (4)$$



where angle brackets ($\langle\rangle$) indicate integration down to the base of the mixed layer. To facilitate interpretation, the mixed layer depth (MLD) is denoted as H in Eq. 4. In this equation, T represents the ocean temperature, u, v, and w denote the zonal,

meridional, and vertical currents, respectively, and $T_H$ is the temperature at the base of the mixed layer.

The MLD is computed using a density threshold criterion of $\Delta\sigma = 0.01$ kg m$^{-3}$, following Gévaudan et al. (2021). This criterion yields MLD values consistent with *in-situ* observations reported by Fernández et al. (2024). In Eq. 4, $Q_s$ represents the solar component of the total heat flux, primarily shortwave radiation (SW). $F_H$ denotes the fraction of solar radiation reaching the ML base. The term $Q_{ns}$ represents the non-solar component of the total heat flux, which can be expressed as:

$$Q_{ns} = \underbrace{\underbrace{DLW}_{\text{Downward longwave radiation}} - \underbrace{ULW}_{\text{Upward longwave radiation}}}_{\text{Radiative heat fluxes}} - \underbrace{\underbrace{LHF}_{\text{Latent heat flux}} - \underbrace{SHF}_{\text{Sensible heat flux}}}_{\text{Turbulent heat fluxes}}. \tag{5}$$

Here, DLW, LHF, and SHF are obtained from WRF, while ULW is computed using Stefan-Boltzmann's law. Note that the flux sign convention differs when computing the MLD heat budget compared to the atmospheric convention: fluxes directed from the ocean to the atmosphere are considered negative, as they contribute to ML cooling. Finally, the EURECA simulation employs the COARE3.0 bulk formulae (Fairall et al., 2003) to compute turbulent heat fluxes (THFs). For consistency, model-

derived THFs are used when calculating the ML heat budget. However, we used COARE3.5 (Edson et al., 2013) to assess LHF sensitivity to SST and surface currents. This does not lead to any inconsistency in the results, since we always compare between LHFs computed with the same algorithm. In addition, we verified that the differences between both algorithms are small compared to the LHF difference values obtained in this article.

Finally, the Residual term in Eq. 4 accounts for horizontal and vertical diffusion, as well as numerical errors associated

with the computation of the other terms. The reader is referred to Section A of the Appendix for further details on the most appropriate way to compute the Residual to minimize numerical errors. Additionally, we explore various criteria to determine whether vertical or horizontal diffusion dominates the Residual in Section B of the Appendix.

Another important quantity used to analyze the effects of water temperature in vertical stratification is the base of the isothermal layer (THERM). Like in Gévaudan et al. (2021), we estimate it as the depth at which the water temperature is

$0.2°C$ lower than the 10 m depth level temperature. Therefore, the barrier layer thickness (BLT) results from the difference between THERM and MLD. Finally, to quantify the relative importance of salinity in ocean stratification, we use the Ocean Stratification Strength (OSS) indicator (Gévaudan et al., 2021; Maes and O'Kane, 2014):

$$OSS = \frac{N_S^2}{N^2}. \tag{6}$$

Here, $N^2$ represents the Brunt-Väisälä frequency, while $N_S^2$ denotes the Brunt-Väisälä frequency computed using a constant

representative temperature, allowing only for salinity variations. OSS values greater than 50% indicate that salinity dominates over temperature in controlling ocean vertical stratification.





## 4 Results

### 4.1 Air-Sea Interface

Fig. 2 presents the JF 2020 mean state of different air-sea interface variables in the Northwest Tropical Atlantic. The shading
in Fig. 2a represents SST, highlighting a warm water stripe ($> 27.3°$C) along the South American coast, extending from the
southernmost coastal point in the domain to Trinidad and Tobago. To its east, a parallel cold SST band ($< 25.5°$C) borders
Trinidad and Tobago and extends northward towards the Lesser Antilles. Further offshore, SST values remain relatively homo-
geneous ( $27°$C), though localized anomalies are observed, such as the warmer region to the east of the Amazon subdomain,
associated with the northward advection of warm Amazon plume waters (not shown). Contours in Fig. 2a show first vertical
level specific humidity, which is highest near the South American coast and decreases towards the northeast.

Fig. 2b displays near-surface wind speed (contours), which varies between 7.5 and 8.2 m s$^{-1}$. The LHF spatial pattern
(shading) closely follows the SST distribution in Fig. 2a, with the highest LHF values ($> 180$ W m$^{-2}$) occurring over the
warm coastal stripe and in the open ocean, and the lowest values ($< 100$ W m$^{-2}$) located over the cold SST band extending
across the continental shelf. Other patches of relatively low LHF are observed offshore over cooler SSTs wuch as the ones west
to Barbados and the Amazon sub-region.

To further characterize the atmospheric JF 2020 spatial pattern, Fig. 2c presents the MABLH (shading) and surface winds
(arrows). Consistent with the DMM mechanism, the shallowest MABL depths ($< 500$ m) align with the cold SST stripe,
whereas deeper MABL values occur over the open ocean, where SSTs and wind speeds are higher. The dominant wind direction
shifts from easterly in the open ocean to northeasterly (trades) closer to the coast. Fig. 2d displays surface currents (arrows) and
SSS (shading). In the southeastern domain, strong northwesterly currents (NBC) advect a low-salinity patch (SSS $< 35$ psu)
associated with the Amazon plume, whose shape and extent are modulated by local eddy-driven circulation. Near Trinidad and
Tobago, a second region of enhanced surface currents exhibits a clockwise rotation, at $10°$N, $58°$W. This eddy, which remains
nearly stationary during JF 2020, lacks strong SSS and SST surface signatures. Such signatures are observed when studying
its vertical structure (not shown).

### 4.2 Coupling Coefficients

To assess the relation between SST/surface current anomalies and the near surface atmosphere at the mesoscale, we compute
the coupling coefficients as defined in Subsection 3.1. Note that these coupling coefficients will be needed to statistically
downscale each of the LHF controlling variables (Eq. 2) in following sections. Since the ocean dynamics is different between
sub-regions, we compute the coupling coefficients separately in EURECA, Amazon, Downstream and Tradewind. Fig. 3,
illustrates the value of the coupling coefficients and the binned linear regressions are displayed in Section C of the Appendix.

The first four salmon markers in Fig. 3 present the spatial distribution of the slope of the linear regression between the
mesoscale surface current vorticity and surface wind curl anomalies. Recall that this linear regression results in the cou-
pling coefficient named $s_w$, used to assess the CFB. The intensity of $s_w$ can be interpreted as the efficiency of the partial
re-energization of the ocean through the wind response to CFB (Renault et al., 2016b, 2019b). Mesoscale $s_w$ values range







**Figure 2.** JF 2020 mean of (a) SST (in °C, shaded) and near-surface specific humidity (in g kg$^{-1}$, contours), (b) LHF (in W m$^{-2}$, shaded) and near-surface wind speed (in m s$^{-1}$, contours), (c) MABLH (in m, shaded) and surface winds (arrows), and (d) SSS (in psu, shaded) and surface currents (in m s$^{-1}$, arrows). In all panels, from north to south, the three white boxes delineate the Tradewind, Downstream, and Amazon sub-regions.

from 0.22 to 0.28, with weaker coupling in the Amazon and Tradewind subdomains (0.22 and 0.24 respectively). The strongest mesoscale coupling is found in EURECA and in the Downstream subdomain, where $s_w$ reaches 0.28 and 0.26 respectively.

In other words, these $s_w$ values indicate that a mesoscale eddy with a velocity of 1 m s$^{-1}$ induces, on average, a wind speed anomaly of 0.28 m s$^{-1}$ in the EURECA domain, which in turn influences LHF estimations through changes in surface winds as shown in Fig. 1b. They also show the spatial variability in the strength of the coupling. The interactions are stronger within the in Downstream than in Tradewind (open ocean).

$s_u$ is displayed in the second four cyan markers of Fig. 3. It is positive in the four domains: warm (cold) SST anomalies increase (decrease) surface winds, which is consistent with the DMM mechanism. $s_u$ is weaker in Tradewind (open ocean,





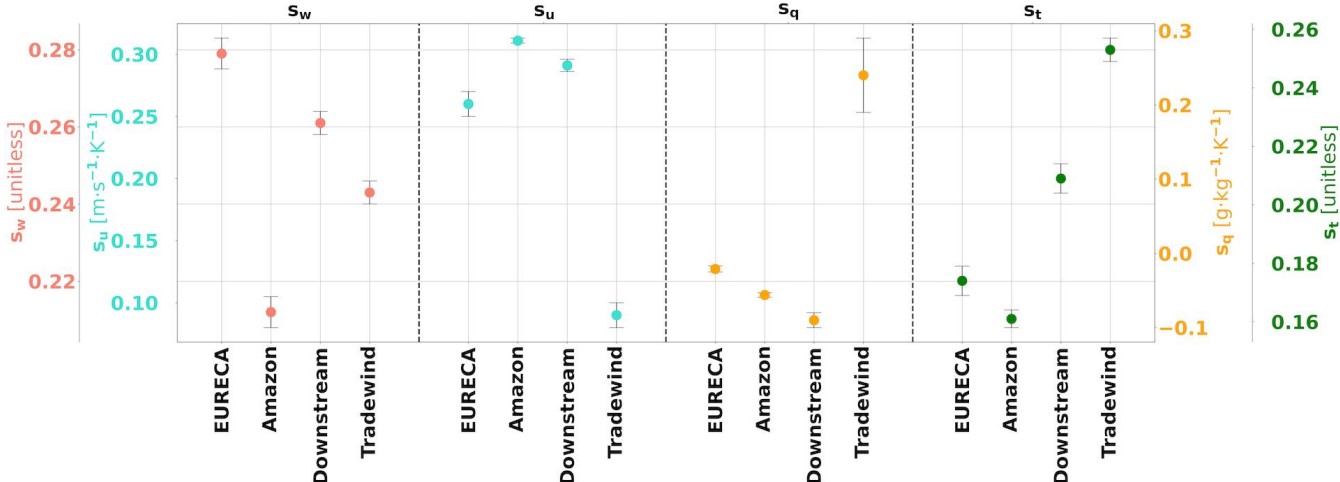

**Figure 3.** Mesoscale coupling coefficients $s_w$ (salmon), $s_u$ (cyan), $s_q$ (orange) and $s_t$ (green). In each dot quartet, from left to right, the markers represent the corresponding coupling coefficient in the EURECA, Amazon, Downstream and Tradewind domains respectively. The error bars depict the standard error of the slope. All the coefficients displayed here are statistically significant at the 95% level after a two-sided $t$-test.

0.11 m s$^{-1}$ K$^{-1}$) than in Amazon and Downstream (warm eddy corridor, 0.29 and 0.31 m s$^{-1}$ K$^{-1}$ respectively). This feature has been previously found in Fernández et al. (2023) using satellite observations. The entire EURECA region produces a $s_u$ of

0.26 m s$^{-1}$ K$^{-1}$ which is in agreement with the January-February-March $s_u$ of Conejero et al. (2024) using the same simulation and hourly data (see their Fig. 9h).

    Another key variable driving LHF is near-surface specific humidity (q). As with wind speed, we use the specific humidity from the first vertical level in the WRF simulation at about 10 m. The four orange markers in Fig. 3 show the associated coupling coefficient $s_q$. It exhibits weak negative values over the Amazon ($-0.05$ g kg$^{-1}$ K$^{-1}$), Downstream ($-0.09$ g kg$^{-1}$ K$^{-1}$) and

EURECA ($-0.03$ g kg$^{-1}$ K$^{-1}$, in agreement with Fernández et al. (2023)), while a stronger positive value appears associated with the Tradewind subdomain (0.24 g kg$^{-1}$ K$^{-1}$). The underlying physical explanation is as follows: at scales larger than the mesoscale, evaporation is sufficient for specific humidity to adjust to SST according to the Clausius-Clapeyron equation. This adjustment rate was estimated to be 1.3 g kg$^{-1}$ K$^{-1}$ through linearization of the Clausius-Clapeyron equation in Fernández et al. (2023). However, at the mesoscale this equilibrium does not hold, as q does not have sufficient time to adjust to SST

variations, resulting in a weak correlation between the two variables. Note that in Amazon and Downstream, the mesoscale $s_q$ is negative meaning that specific humidity decreases with increasing SST. This behavior may reflect the entrainment of colder and drier air from the free troposphere due to the DMM mechanism. We will come back to this point later.

    Finally, the last four green markers in Fig. 3 represent $s_t$ in the different domains. As expected, they all are positive: higher SST implies higher surface air temperature. In addition, $s_t$ is strongest in Tradewind ($\sim$0.25) where the SST-associated wind





speed and specific humidity variations are weaker. This might result from a decreased advection of other air masses with other

temperatures which could locally modify temperature values close to the surface.

### 4.3   LHF Sensitivity to SST and Currents

To evaluate the representation of LHF sensitivity to SST in the EURECA simulation, we perform a linear regression between

LHF variations and the SST correction ($\Delta$SST, see Eq. 3) for the different LHF subsets detailed in Subsection 3.2. The results

are presented in Fig. 4 for the EURECA domain (first row), the Amazon sub-region (second row), the Downstream sub-region

(third row), and the Tradewind sub-region (fourth row).

    The estimated LHF change per °C associated with the presence of the mesoscale ocean is shown in orange in Figs. 4a,

c, e, and g, representing the difference between LHF$_{\mathrm{HR}}$ and LHF$_{\mathrm{LR}}$. Recall that LHF$_{\mathrm{HR}}$ is computed using the statistically

downscaled WRF variables (Eq. 2) in COARE3.5, while LHF$_{\mathrm{LR}}$ corresponds to LHF derived from the *smoothed* variables,

obtained by subtracting low-pass filtered values (Gaussian filter with a 250 km cutoff length) from the raw data.

    The mean LHF sensitivity to mesoscale SST in the EURECA region is 47.7 W m$^{-2}$ K$^{-1}$ and the Amazon and Downstream

sub-regions show similar values (46.7 W m$^{-2}$ K$^{-1}$ and 46.9 W m$^{-2}$ K$^{-1}$ respectively). On the contrary, the sensitivity is

lower in the Tradewind subdomain (35 W m$^{-2}$ K$^{-1}$). Given the mean LHF values of 150 W m$^{-2}$ in EURECA, 130 W m$^{-2}$

in Amazon, 170 W m$^{-2}$ in Downstream, and 140 W m$^{-2}$ in Tradewind, the slopes of the linear regressions correspond to

31.8 % K$^{-1}$ (47.7/150), 35.9 % K$^{-1}$ (46.7/130), 27.5 % K$^{-1}$ (46.9/170), and 25 % K$^{-1}$ (35/140), respectively. These values

align with the theoretical estimate of approximately 33 % K$^{-1}$ of Fernández et al. (2023). In addition, they are consistent

with satellite (33.8 % K$^{-1}$) and reanalysis (26.6 % K$^{-1}$) estimates obtained in Fernández et al. (2023). The observed regional

differences were also reported in that study, with stronger LHF variations near the South American coast (Amazon and Down-

stream) compared to the open ocean (Tradewind). Finally, Conejero et al. (2024), using the same simulation, computed the

sensitivity of THF mesoscale anomalies with respect to SST mesoscale anomalies. They obtained a value of 53 W m$^{-2}$ K$^{-1}$

in January-February-March for the whole EURECA region, slightly larger than our 47.7 W m$^{-2}$ K$^{-1}$ (see their Fig. 10). This

is partly because of the effect of sensible heat flux (present in the THF) which adds to the LHF sensitivity to SST and partly

due to the fact that we consider JF.

    We compare the LHF sensitivity results with *in-situ* observations as well. In Fernández et al. (2024), the LHF changes

were assessed across mesoscale SST anomalies of 2°C and $-0.4$°C, in the Amazon and Downstream sub-regions, respec-

tively. The first SST anomaly induced a LHF difference of approximately 160 W m$^{-2}$ between itself and its environment

($\sim$80 W m$^{-2}$ K$^{-1}$). The second SST anomaly, resulted in a LHF difference of 95 W m$^{-2}$ ($\sim$38 W m$^{-2}$ K$^{-1}$). Therefore, the

latter agrees better with the model estimate for the Downstream subdomain, while the former is significantly larger, likely due

to its proximity to the coast. Indeed, the strongest SST gradients in the model also appear over the continental shelf (Fig. 2a)

and are not included in this analysis. We should also keep in mind that LHF sensitivity to SST across fronts in *in-situ* ob-

servations is subject to several uncertainty sources which could modify its value and make the direct comparison with model

estimates less straightforward. The relative orientation between the sampling device's trajectory and the SST front or even just

defining the front's location are amont these uncertainty sources.





**Figure 4.** LHF sensitivity to $\Delta$SST (see Eq. 3) across different regions. Each row corresponds to a specific domain: the EURECA region (first row), the Amazon sub-region (second row), the Downstream sub-region (third row), and the Tradewind sub-region (fourth row). Panels (a), (c), (e), and (g) display the difference between $LHF_{HR}$ and $LHF_{LR}$ as a function of $\Delta$SST. Panels (b), (d), (f), and (h) show the differences between $LHF_{therm}$ and $LHF_{LR}$, and between $LHF_{therm-U}$ and $LHF_{LR}$, as a function of $\Delta$SST (blue and orange, respectively). The methodology used to compute these LHF datasets is detailed in the main text. The SST anomaly values are binned into 2% percentile intervals, ensuring each contains more than 15 000 values. Vertical error bars indicate the standard deviation from the mean in each interval, while straight lines represent the least-squares regression fits. The regression slope $\pm$ standard error (p-value) is reported at the bottom of each panel.





To quantify the thermodynamic contribution to LHF sensitivity (i.e. the component linked only to SST changes, denoted as
LHF$_{\text{therm}}$), we compute its linear regression shown in dark blue in Figs. 4b, d, f, and h. In the EURECA domain, this contribution accounts for a LHF change of 6.6% K$^{-1}$ (9.9/150). In the other sub-regions, it represents 6 % K$^{-1}$ (Amazon, 7.76/130),
4.9 % K$^{-1}$ (Downstream, 8.4/170) and 6.7 % K$^{-1}$ (Tradewind, 9.5/140). The increasing importance of the thermodynamic
contribution towards the open ocean is consistent with the particular air-sea coupling in these regions (lower $s_u$ and $s_t$ and
larger $s_q$, see Fig. 3). Overall, these LHF sensitivity values show that in the presence of mesoscale anomalies, LHF variation at
the mesoscale are mainly associated with the dynamic contribution (i.e LHF variations linked to the SST-induced modification
of the near-surface atmospheric variables) whereas the thermodynamic contribution remains a second-order contributor, in
agreement with previous observation and reanalysis-based studies (Fernández et al., 2023).

For completeness, we also compute LHF$_{\text{therm}-U}$ − LHF$_{\text{LR}}$ as a function of ΔSST, shown in orange in Figs. 4b, d, f, and
h. This allows us to separate the surface wind contribution from the specific humidity's within the dynamic component of
LHF sensitivity. In EURECA, Amazon and Downstream the effect of surface wind variations adds around 8 W m$^{-2}$ K$^{-1}$
to the thermodynamic contribution. Regarding Tradewind, the wind speed contribution is more modest, representing only an
additional 4.7 W m$^{-2}$ K$^{-1}$. Therefore, within the dynamic contribution, the majority of the LHF change is linked to the weak
mesoscale SST-specific humdity coupling. The air temperature effect is even smaller than the wind speed and it is not shown
here for the sake of simplicity.

To quantify the impact of using relative winds (i.e., surface winds minus currents) in LHF computations, we perform a
linear regression between LHF$_{\text{orig}}$ − LHF$_{\text{U}}$ and the difference between the norms of relative winds and surface winds. Here,
LHF$_{\text{U}}$ represents the dataset computed using only the closest-to-the-surface winds (without currents) and LHF$_{\text{orig}}$ stands for
the LHF computed using relative winds. The regressions are shown in dark blue in Fig. 5 for the EURECA (a), Amazon (b),
Downstream (c), and Tradewind (d) regions.

Across the EURECA domain, surface wind speed generally exceeds relative wind speed. Winds predominantly blow towards
the southwest ($-110°$ from north), while surface currents are oriented northwestward ($-70°$ from north). As a result, considering only surface winds increases LHF by up to 10 W m$^{-2}$, a trend observed across all sub-regions. The sign and magnitude
of LHF$_{\text{orig}}$ − LHF$_{\text{U}}$ depend on the relative orientation of winds and currents. In the Downstream sub-region (Fig. 5c), the
southern edge of an anticyclonic eddy (Fig. 2d) aligns surface currents with surface winds ($-120°$ from the north), leading to
$|\overrightarrow{U}| > |\overrightarrow{U}\text{-}\overrightarrow{U_o}|$ and thus reducing LHF when relative winds are used. On the contrary, Amazon and Tradewind (Fig. 5b and d)
exhibit both positive and negative LHF differences, consistent with its broader wind and surface current distributions (Figs. 2c
and d). These LHF variations align well with observations. Using the same representative LHF values for each sub-region,
we find that the regression slopes correspond to 10.9, 11.3, 9.4, and 13 % K$^{-1}$ in the EURECA, Amazon, Downstream, and
Tradewind regions, respectively. *In-situ* observations report a sensitivity range of 5 % K$^{-1}$ to 15 % K$^{-1}$ (Fernández et al.,
2024).

Finally, to isolate the CFB contribution to LHF variations, we perform a linear regression between LHF$_{\text{orig}}$ − LHF$_{\text{no}-\text{CFB}}$
and the difference between relative wind norms with and without current feedback. The results, shown in orange in Fig. 5,
exhibit a similar sensitivity than the regression concerning LHF$_{\text{orig}}$ − LHF$_{\text{U}}$. Whereas $|\overrightarrow{U}\text{-}\overrightarrow{U_o}|$ is typically smaller than $|\overrightarrow{U}|$,





**Figure 5.** LHF sensitivity to surface currents for (a) EURECA, (b) Amazon, (c) Downstream, and (d) Tradewind. In all panels, blue markers represent the difference between $\text{LHF}_\text{U}$ and $\text{LHF}_\text{orig}$ as a function of the difference between the norm of the first vertical level wind velocity and the relative wind velocity field. Orange markers indicate the difference between $\text{LHF}_\text{orig}$ and $\text{LHF}_\text{no-CFB}$ as a function of the difference between the norms of relative winds, with and without the CFB effect. The methodology used to derive these LHF datasets is detailed in the main text. The x-axis variable is divided into intervals containing an equal number of values, using a 2% percentile separation, ensuring more than 15 000 values per interval. Vertical error bars indicate the standard deviation relative to the mean for each interval, while straight lines denote the least-squares regression fits. The regression slope ± standard error (p-value) is displayed in each panel's legend.

the relative velocity in presence of CFB is typically larger than the relative velocity without CFB. In addition, the CFB effect is
one order-of-magnitude smaller (within ±3 W m⁻²), consistent with *in-situ* results from Fernández et al. (2024). According to
CFB, surface currents generate wind anomalies in their own direction. Thus, in the Downstream region, where surface currents
mostly align with wind direction, CFB enhances relative winds and LHF. In regions without a dominant current direction, LHF
variations can be either positive or negative.





## 4.4    Vertical Ocean and Atmosphere Structure in Amazon

To fully explain the modeled LHF variations, we examine the atmospheric response and oceanic configuration associated with mesoscale SST anomalies. Here, we focus on the Amazon sub-region, since we aim to link the mesoscale SST anomalies to the presence of the Amazon plume and its heat budget. Results for the Downstream and Tradewind sub-regions and EURECA yield similar conclusions in terms of physical mechanisms at play and for brevity we do not include them in this study. The analysis follows what is presented in Borgnino et al. (2025) for an atmosphere-only model forced with realistic SSTs.

Fig. 6 presents the binned distribution of the marine atmospheric boundary layer (MABL) vertical structure, as well as the dependence of SST, SSS, and mixed layer depth (MLD) on the SST mesoscale anomaly. Over cold SST anomalies, the MABL exhibits increased stability, characterized by positive values of the Brunt-Väisälä frequency anomalies ($N^2$), particularly above the MABL height (black line in Fig. 6a). Conversely, over warm SST anomalies, the atmosphere becomes more unstable, with negative values of $N^2$.

Changes in atmospheric vertical stratification also influence wind speed variations (Fig. 6b). Two distinct anomaly dipoles are observed at the warmest and coldest ends of the histogram. Over the highest SST anomalies, wind speed increases slightly near the surface (around 0.05 m s$^{-1}$), while negative wind speed anomalies dominate above. This behavior aligns with the downward momentum mixing (DMM) mechanism: downward fluxes of momentum from the free troposphere reduce wind speed at higher levels while enhancing it near the surface. The opposite pattern is observed over the coldest SST anomalies in the Amazon sub-region, where wind speed decreases near the surface and increases at and above the MABL height.

Furthermore, the warming (cooling) induced by mesoscale SST anomalies extends well above the MABL height (MABLH) (Fig. 6c). This is reflected in the two-dimensional saturation specific humidity anomaly histogram, where values exceed $\pm 0.1$ g kg$^{-1}$ K$^{-1}$ in the warmest (coldest) intervals. In contrast, specific humidity variations are weaker, as suggested by the Amazon ($s_q$). Consistent with the Amazon $s_q$ shown in Fig. 3, q slightly decreases over the warmest mesoscale SST anomalies and slightly increases in the MABL located over the coldest SST anomalies. At the MABLH level, we find slightly positive q anomalies which then become negative further above, except for the warmest mesoscale SST anomalies. This pattern of specific humidity anomalies aligns again with the DMM: warmer SSTs destabilize the MABL leading to the entrainment of drier (and colder) air from the free troposphere towards the surface, decreasing surface q (and air temperature, thus reducing $s_t$ with respect to Tradewind where $s_q$ is positive as shown in Fig. 3). Consequently, the specific humidity deficit ($\Delta q$) anomalies (Fig. 6e) mostly resemble those of the saturation specific humidity ($q_s$) in Fig. 6c, with negative $\Delta q$ anomalies coinciding with negative SST anomalies, and vice versa. The only exception corresponds to the warmest mesoscale SST anomalies where negative anomalies of $\Delta q$ emerge over the MABLH associated with the reduced q.

Fig. 6f presents the cloud water mixing ratio (QCLOUD). Although anomalies remain weak, an increase in QCLOUD is observed at the MABLH level for the majority of the mesoscale SST anomaly range. In regions of negative mesoscale SST anomalies, the strongest QCLOUD increases are confined close to the MABLH. This finding aligns with previous observational studies, such as Acquistapace et al. (2022), which detected stratiform shallow clouds over stable tropical MABLs, and mod-







**Figure 6.** Panels (a)–(f) show the vertical structure of the first 2000 m of the atmosphere (in terms of mesoscale anomalies) as a function of the SST mesoscale anomaly: (a) Brunt-Väisälä frequency, (b) horizontal wind speed, (c) saturation specific humidity, (d) specific humidity, (e) specific humidity deficit, and (f) cloud water mixing ratio (QCLOUD). In all these panels, the black line represents the MABL height (MABLH), while the purple line indicates LHF. Panel (g) displays SST (red) and SSS (blue), and panel (h) shows MLD, both as functions of the SST mesoscale anomaly. All calculations are based on daily averages of the model output for JF 2020. Bins are defined using the 2% percentile range to ensure a sufficient number of values per bin.





eling studies (Borgnino et al., 2025). Over positive mesoscale SST anomalies, QCLOUD positive anomalies extend upward potentially yielding more vertically developed clouds in the context of a more unstable atmosphere.

In all the panels discussed above, the purple line represents the mean binned LHF as a function of the SST mesoscale anomaly. Consistent with observational results from both satellite (Fernández et al., 2023) and *in-situ* data (Fernández et al., 2024), a significant LHF increase of approximately 50 W m$^{-2}$ is observed. This increase is driven by the combined effects of SST changes (thermodynamic contribution), as well as variations in specific humidity and, to a lower extent, wind speed (dynamic contribution), which were quantified separately in the previous section. Additionally, the black line represents the simulated MABLH mean per mesoscale SST anomaly bin. Its values remain around 600 m for most SST anomaly bins, consistent with expected tropical ocean conditions. A slight increase in MABLH is observed toward the warmest SSTs; however, this increase is notably weaker than that documented in *in-situ* observations by Fernández et al. (2024). Thus, the MABL response does not fully align with the downward momentum mixing (DMM) mechanism, although changes in vertical stratification, wind speed and specific humidity suggest that DMM is active at the mesoscale in the Amazon sub-region. We hypothesize that this discrepancy may arise from the MABL stability parameterizations used within the model. A more detailed analysis would be required to confirm this hypothesis, but it is beyond the scope of this study.

Note that all the anomalies (atmospheric and SST) shown in Fig. 6 are small compared to the ones shown in Borgnino et al. (2025). A direct comparison is not straightforward since their simulation is SST-forced and ours is coupled, and they use a different filtering procedure: no time filter and a high-pass Gaussian filter with a 150 km cutoff length. In addition, they evaluate the vertical atmospheric structure of the whole EURECA domain (not only Amazon). However, in the range of the anomalies presented in this article ($\pm 0.3$°C), the results are consistent. For example, in their Fig. 4d wind speed mesoscale anomalies range from $-0.2$ m s$^{-1}$ to 0.2 m s$^{-1}$ and SST mesoscale anomalies from $-0.6$°C to 0.5°C. Nevertheless, between $\pm 0.3$°C their wind speed anomaly values lie within the $\pm 0.05$ m s$^{-1}$ range of Fig. 6b.

Finally, Figs. 6g and h illustrate the dependence of SST, SSS, and MLD on SST anomalies. SSS peaks at a mesoscale SST anomaly of $-0.15$°C, corresponding to the most open-ocean region of the Amazon domain. In contrast, the core of the Amazon plume, where salinity is lowest, corresponds to the warmer waters (a mesoscale SST anomaly of 0.1°C). Additionally, a relatively low SSS ($\sim$35.5 psu) is observed in the SST mesoscale anomaly range associated with the cold stripe waters mixing with the Amazon plume (SST mesoscale anomalies around $-0.25$°C). The MLD is shallowest in the core of the plume (SST mesoscale anomaly of $\sim$0.1°C) and deepens toward the open ocean (between 0 and $-0.1$°C of SST mesoscale anomaly). Coastal waters associated with the cold SST stripe exhibit intermediate MLD values, not reaching 40 m.

We now analyze the rest of the terms of the surface heat balance (Eq. 5) in Figs. 7a and b as a function of the SST mesoscale anomaly. Here, the sign convention is inverted: positive fluxes indicate heat transfer from the atmosphere to the ocean, and vice versa. Fig. 7a reveals that the net heat flux decreases from over 50 W m$^{-2}$ at $-0.3$°C SST mesoscale anomalies to negative values for anomalies above 0.2°C. This decrease is primarily driven by LHF variations and, to a lesser extent, by changes in the sensible heat flux (SHF), which declines from slightly above 0 W m$^{-2}$ in the coldest regions to $-10$ W m$^{-2}$ in the warmest bins. Note that the net radiative heat flux variations' (orange line) are smaller than those associated with LHF.



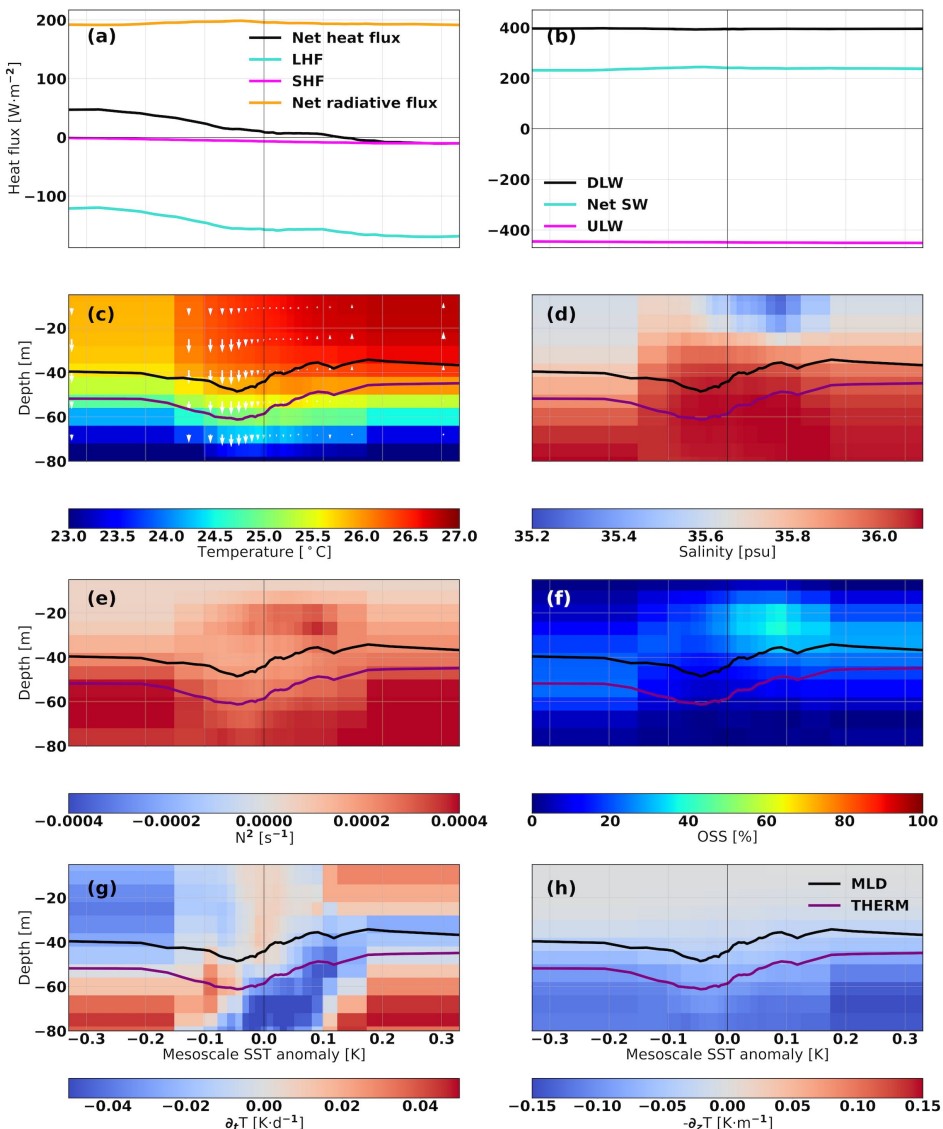

**Figure 7.** Behavior of various ocean properties as a function of the SST mesoscale anomaly. In (a), the black line represents the net heat flux, the turquoise line corresponds to LHF, the magenta line to SHF and the orange line the net radiative heat flux. In (b), the black line denotes downward longwave radiation, the turquoise line indicates net shortwave radiation, and the magenta line represents upward longwave radiation. Note that the sum of these three contributions yields the net radiative flux represented in orange in panel a. In both panels, the sign convention is such that positive fluxes are directed toward the ocean. Panels (c)–(h) illustrate the vertical structure of the upper 120 m of the ocean: (c) water temperature, (d) salinity, (e) Brunt-Väisälä frequency, (f) OSS index, (g) total temperature tendency, and (h) vertical temperature gradient (a negative gradient indicates decreasing temperature with depth). In these panels, the black line represents the MLD, while the purple line denotes the base of the isothermal layer. All values are based on daily model output averages for JF 2020. As in previous figures, SST anomaly bins are defined using the 2% percentile separation to ensure a sufficient number of values per bin.



We decompose the radiative heat flux contribution into its components (downward longwave radiation, shortwave radiation, and upward longwave radiation). The net shortwave heat flux remains nearly constant (cyan line in Fig. 7b), indicating a negligible impact of MABLH cloud cover changes (Fig. 6f) on air-sea heat fluxes in the Amazon sub-region. Consequently, net heat flux variations are primarily driven by turbulent heat flux (THF) modulations.

To gain deeper insight into the processes that govern heat redistribution within the ocean interior and their relationship with mesoscale SST anomalies, we examine the two-dimensional histograms (depth versus mesoscale SST anomaly) of several key variables in Figs. 7c–h. The warmest mesoscale anomaly SST intervals correspond to water temperatures exceeding 27°C down to the MLD, which remains around 35 m. These warmer waters are associated with low salinity values, confirming that the strongest positive mesoscale anomalies correspond to the core of the Amazon plume. Notably, enhanced vertical stratification

within the Amazon plume, linked to elevated Brunt-Väisälä frequency values over low-salinity waters (Fig. 7e), coincides with weak vertical motion (white arrows in Fig. 7c). Indeed, the OSS index, which quantifies the fraction of stratification attributed to salinity, indicates that salinity plays a more important role in vertical stratification. Over the core of the plume (lowest salinity and $-0.1$°C of mesoscale SST anomaly), OSS values reach 50%.

In contrast, the saltiest MLDs correspond to stronger vertical velocities, occurring around the mesoscale anomaly $-0.1$°C.

The coldest waters ($<-0.3$°C) also exhibit relatively low salinity, a signal already observed in SSS (Fig. 6g), extending throughout the ML. Despite their lower temperature, reduced salinity results in shallower MLs, weaker vertical motion, and increased stratification, particularly due to salinity effects. Fig. 7g presents the binned distribution of total temperature tendency as a function of mesoscale SST anomaly. The temperature tendency within the ML is negative for the coldest SST anomalies and positive for the Amazon plume-related bins (higher mesoscale SST anomalies), where a warming trend is observed

despite the net heat flux indicating heat loss from the ocean to the atmosphere (Fig. 7a). We return to this point later. Below the plume-related ML, cooling is evident, consistent with the high turbidity of plume-related waters rich in chlorophyll and nutrients (Olivier et al., 2022), which significantly reduces the amount of solar radiation reaching the MLD.

To investigate the presence of temperature inversions (observed in *in-situ* work, (Fernández et al., 2024)) and potential subsurface heat release, Fig. 7h presents the two-dimensional histogram of the vertical temperature gradient ($\partial_z T$). In general,

$\partial_z T$ peaks below the base of THERM (the isothermal layer computed as the depth where temperature is 0.2°C colder than SST). However, in all cases, $\partial_z T$ remains negative below the MLD, indicating a sharp decrease in temperature with depth and the absence of temperature inversions. Thus, unlike the *in-situ* observations analyzed by Fernández et al. (2024), the model fails to reproduce the temperature inversions occurring between the MLD and THERM. Consequently, their contribution to LHF spatial variability cannot be isolated.

To further investigate heat transfer in and out of the ML, particularly to explain why the ML undergoes net warming over plume waters despite losing heat to the atmosphere, Fig. 8 displays the various terms of the ML heat budget (Eq. 4) averaged over JF 2020. Additionally, we present the spatial distribution of MLD and the barrier layer thickness (BLT) with superimposed SST contours in Figs. 8a and b, respectively. The MLD map clearly distinguishes three water masses within the Amazon subdomain. To the west, cold waters mix with the warm and fresh Amazon plume, triggering a decoupling between haline and







**Figure 8.** Averaged JF 2020 mixed layer heat budget (vertically integrated down to the mixed layer depth) for the Amazon sub-region. (a) Mean MLD and (b) mean barrier layer thickness (BLT), with black contours representing SST. Panels (c)–(h) illustrate the different terms in Eq. 4: (c) total temperature tendency, (d) horizontal advection, (e) vertical advection, (f) entrainment, (g) atmospheric forcing, and (h) residual. The hatched areas in (f) indicate locations where the residual is primarily associated with vertical diffusion, following Cronin et al. (2015).



thermal stratification and leading to the formation of barrier layers (BL) as thick as 35 m. In the eastern part of the domain, relatively cold and saline waters exhibit the deepest MLDs, exceeding 50 m.

The JF 2020 averaged total temperature tendency ($\partial_t T$) exhibits a highly heterogeneous pattern (Fig. 8a). In the southwestern part of the domain, warming is primarily driven by the horizontal advection of warmer waters from the south (Fig. 8b).

In contrast, the Amazon plume-related waters display a cool-warm dipole. The waters with the lowest salinity (SSS <
35 psu) experience net cooling, mainly due to atmospheric forcing (Fig. 8g). However, the warmest waters (SST > 26.7°C) with salinity values between 35 and 35.4 psu exhibit positive $\partial_t T$. This increase results primarily from horizontal advection, as the plume detaches from the continental shelf, and warm core waters are transported northwestward, leading to a local temperature increase. The nearly uniform temperature within the plume explains the absence of strong horizontal advection values in its interior, as warm water from the surroundings is advected into a region where little temperature variation occurs.

In addition to horizontal advection, entrainment contributes to the temperature tendency. Positive entrainment values are found along the 35.5 psu isoline (Fig. 8f), likely resulting from MLD variations associated with the lateral displacement of the plume. However, the magnitude of this contribution is an order of magnitude weaker than that of lateral advection. A third contributing factor is vertical diffusion. Positive values of the mixed layer heat budget residual appear around the 35.5 psu isoline (Fig. 8h). This residual can primarily be attributed to vertical diffusion, as it meets the Cronin et al. (2015) criteria used
in this study (detailed in Section B of the Appendix), which include the presence of a strong vertical temperature gradient, relatively weak horizontal advection at the MLD, and the absence of a barrier layer (BL, the difference between THERM and the MLD). The impact of vertical diffusion is comparable in magnitude to lateral advection. The combined effects of horizontal advection, entrainment, and vertical diffusion outweigh the cooling effect of the atmospheric forcing (Fig. 8g).

## 5   Discussion and Conclusion

High-resolution coupled simulations serve as a powerful complement to remote sensing, reanalysis, and *in-situ* observations to study ocean mesoscale influences on latent heat flux (LHF) variability in the Northwest Tropical Atlantic. In this study, we employ the WRF-CROCO coupled EURECA simulation at 1 km oceanic and 2 km atmospheric resolution, fully resolving ocean mesoscale ($O$(50 - 250 km)) processes. We focus on January–February 2020, when the Intertropical Convergence Zone (ITCZ) shifts southward, and analyze four domains: the full EURECA region; Amazon and Downstream (coastal regions, with
the former influenced by the Amazon plume); and Tradewind (a more quiescent open-ocean region).

Our analysis of air-sea coupling coefficients confirms a robust mesoscale positive correlation between surface current vorticity and wind curl across all regions. The associated coupling coefficient is denoted as $s_w$. This correlation is the strongest in the Downstream sub-region ($s_w = 0.26$, Fig. 3e) compared toTradewind ($s_w = 0.24$) and Amazon ($s_w = 0.21$). Although these variations might seem small, the error bars associated with these coupling coefficients do not overlap (Fig. 3, salmon
markers) and all $s_w$ values are statistically significant (see Section C of the Appendix). They represent changes of 14.3% in $s_w$ (i.e. (0.26-0.22 / 0.28)). This difference might be related to the presence of stronger surface currents in Downstream than in





Amazon and Tradewind associated with the nearly-stationary eddy located in front of Trinidad and Tobago (Fig. 2d). Stronger currents may drive a more robust wind response, less likely to be masked by other atmospheric processes.

Most coupling coefficients at the mesoscale are weaker than satellite-derived values in Fernández et al. (2023) but agree
with those reported in Renault et al. (2019b). We find $s_u = 0.26$ m s$^{-1}$ K$^{-1}$ for the EURECA domain, peaking in Amazon and Downstream (0.29 and 0.31 m s$^{-1}$ K$^{-1}$ respectively, Fig. 3) and reaching its minimum in Tradewind (0.11 m s$^{-1}$ K$^{-1}$, Fig. 3). This pattern reflects stronger Downward Momentum Mixing (DMM) activity in Amazon and Downstream than in Tradewind. Meanwhile, $s_q$ is slightly negative in and Amazon Downstream ($-0.05$ and $-0.09$ g kg$^{-1}$ K$^{-1}$ respectively) and positive in Tradewind (0.24 g kg$^{-1}$ K$^{-1}$, Fig. 3). Finally, $s_t$ is always positive and ranges from 0.16 in Amazon (in the warm eddy
corridor) to 0.25 in Tradewind (open ocean).

We quantify LHF sensitivity to sea-surface temperature (SST) anomalies through linear regression analysis (Fig. 4). Consistent with satellite observations (Fernández et al., 2023), LHF increases by 31.8% per 1°C warming, with the strongest sensitivity in Amazon (35.9% K$^{-1}$, Fig. 4c) and the weakest in Tradewind (25% K$^{-1}$, Fig. 4g). Furthermore, we separate the thermodynamic (SST-driven) and the dynamic (atmospheric response) contributions, confirming that dynamic effects domi-
nate. The thermodynamic contribution represents only 4.5%–7% per 1°C, with specific humidity driving most dynamic LHF changes. Finally, we assess LHF sensitivity to surface currents (Fig. 5). Accounting for relative winds instead of absolute winds induces LHF variations up to 15 W m$^{-2}$, consistent with *in-situ* results (Fernández et al., 2024). The current feedback (CFB) effect is much smaller, contributing at most 3 W m$^{-2}$.

The vertical structures of the marine atmospheric boundary layer and mixed layer reveal key mechanisms behind the ob-
served coupling. In the atmosphere, we observe DMM-consistent patterns, with a dipole in $N^2$ (more unstable over warm SSTs, more stable over cold SSTs) and a reduction (increase) of near-surface specific humidity (near-surface winds) over warm SST anomalies. Similar findings hold for Downstream and Tradewind (not shown). In the ocean, the warmest SST anomalies coincide with the Amazon plume core (Figs. 7c and d). They are characterized by shallower MLDs, inhibited water vertical velocity and a negative net heat flux towards the atmosphere. Despite this, positive mesoscale anomalies associated with the Amazon
plume tend to continue warming ($\partial_t T > 0$) mainly because of lateral advection of warm plume-related waters, as confirmed by the mixed layer heat budget (Fig. 8).

This study extends the mesoscale findings of Fernández et al. (2023, 2024) providing a regionalized analysis of the air-sea coupling and the ocean-atmosphere vertical structure. It also links the air-sea coupling with the Amazon plume. However, our analysis is restricted to boreal winter (January and February). Future work should investigate other seasons in the EURECA
simulation (June 2019–June 2020), as coupling coefficients vary seasonally (Conejero et al., 2024), with stronger air-sea interactions in boreal winter than in summer. Additionally, the Amazon runoff peaks in summer, amplifying the Amazon plume's influence on LHF heterogeneity. Moreover, this study does not address the ocean submesoscale ($O(<50$ km$)$) although evidence suggests that submesoscale ocean structures also impact the near-surface atmosphere (Meroni et al., 2018; Gaube et al., 2019). This is mainly because daily averaging, used to remove the diurnal cycle, might also eliminate important submesoscale
signals. A more sophisticated approach, such as multichannel singular spectrum analysis (M-SSA), could better preserve submesoscale variability. Furthermore, comparing the higher-resolution EURECA simulation (1 km ocean, 2 km atmosphere) with



the coarser Antilles simulation (2.5 km ocean, 6 km atmosphere) would clarify the impact of spatial resolution on mesoscale and submesoscale air-sea interactions. Finally, longer simulations ($\geq$10 years) would enable interannual variability studies, particularly regarding Amazon plume detachment, which occurs irregularly (e.g., absent in 2010, 2011, and 2013; (Olivier et al., 2022)).

Our findings emphasize the need for high-resolution modeling in climate studies. Traditional climate models rely on coarse SST grids, which suppress the small-scale air-sea disequilibrium that governs LHF release. Implementing the LHF downscaling algorithm from Fernández et al. (2023) in model couplers could enhance air-sea flux representation in high-resolution climate simulations. Future work should explore its implementation within global coupled models to improve energy exchange parameterizations.

*Data availability.* Apart from the EURECA simulation, we benefited from several data sets made freely available and listed here.

- SeaFlux (Roberts et al., 2020), http://dx.doi.org/10.5067/SEAFLUX/DATA101

- SMAP maps produced by Remote sensing systems (RSS v4 40 km) (Boutin et al., 2021),
  https://doi.org/10.5285/5920a2c77e3c45339477acd31ce62c3c

- Global Ocean Gridded L4 Sea Surface Heights And Derived Variables Reprocessed 1993 Ongoing (CLS, 2018), https://doi.org/10.48670/moi-00148

*Code and data availability.* All the codes used to produce the figures of this article are available upon request to the first author.





## Appendix A: Numerical Considerations in the Mixed Layer Heat Budget Computation

In the EURECA simulation, only the monthly means of the advective-diffusive equation terms are stored. For temperature, this
equation reads:

$$\underbrace{\partial_t \mathrm{T}}_{\text{Total tendency}} + \underbrace{\nabla \cdot (\vec{\mathrm{u}}\, \mathrm{T})}_{\text{Advection}} = \underbrace{\mathcal{F}}_{\text{Forcing}} + \underbrace{\mathcal{D}}_{\text{Mixing}} . \qquad (A1)$$

One of the challenges in computing the mixed layer (ML) heat budget is ensuring its closure, as numerical choices in the calculation of each term can introduce discrepancies. To mitigate this issue, we followed the computational steps detailed in the CROCO documentation when computing all the terms in Eq. A1 on a daily basis.

Fig. A1 presents a comparison between the recomputed and stored Eq. A1 terms, integrated down to 100 m (a depth beyond any mixed layer depth to minimize entrainment effects). The left column (except for the last row) displays the recomputed terms, the central column shows the corresponding stored values from the simulation output, and the right column represents the numerical bias, i.e., the difference between the recomputed and stored values.

The total temperature tendency, averaged over February 2020, exhibits a consistent spatial pattern between the recomputed
(Fig. A1a) and stored (Fig. A1b) values. A region of strong cooling extends from the southern part of the domain (around 54°W) towards the north and northwest, surrounded by positive temperature tendency values, particularly in the southernmost part of the domain. The numerical bias (Fig. A1c) is one to two orders of magnitude smaller than the actual temperature tendency values, indicating that the numerical error in recomputing the total tendency is relatively small.

However, significant differences arise when comparing the advection terms. The recomputed horizontal (Fig. A1d) and
vertical (Fig. A1g) advections differ substantially in both pattern and intensity from their stored counterparts (Figs. A1e and h, respectively). The recomputed advections are systematically stronger, and their differences (Figs. A1f and i) are of the same order of magnitude as, or even exceed, the values themselves.

In contrast, the atmospheric forcing term remains consistent between the recomputed (Fig. A1j) and stored (Fig. A1k) datasets, with negligible numerical bias (Fig. A1l). This suggests that numerical errors primarily impact the advection terms,
while the atmospheric forcing is well represented in the recomputed dataset.

The discrepancy in the advection terms leads to a significantly large residual field (Fig. A1m), which exceeds the total temperature tendency itself (Fig. A1a). This stands in contrast to the stored residual field (Fig. A1n), where significant values are only observed around 7.5°N. The difference between the recomputed and stored residual fields (Fig. A1o) is almost as large as the recomputed residual field itself, highlighting the impact of numerical discrepancies. Additionally, the stored residual field
is evenly partitioned between horizontal and vertical mixing, whereas these two contributions cannot be separately estimated from the recomputed fields.

Overall, Fig. A1 underscores the critical role of numerical schemes in computing heat budgets, particularly for advection and derivative calculations. In this case, the recomputed horizontal and vertical advections were obtained using a second-order centered scheme, whereas the simulation employs a fifth-order upstream advection scheme, leading to significant differences in





**Figure A1.** Comparison between the stored advective-diffusive equation terms (Eq. A1 and the recomputed ones, integrated over a layer of 100 m, (deeper than the maximum MLD recorded).





the resulting fields. Thus, in the mixed-layer heat budget-related results, we will work with the stored advection terms whereas
the rest of the terms (total tendency, entrainment and atmospheric forcing) will be recomputed each day from model data and
then averaged in a monthly basis.

**Appendix B: Criteria to Associate the Residual from the Mixed Layer Heat Budget to Vertical Diffusion**

A common approach to estimating vertical heat diffusivity is to infer it from the residual of the mixed layer heat budget

(Girishkumar et al., 2020). Cronin et al. (2015) extended this approach by proposing selection criteria for estimating vertical heat diffusivity at the ML base. First, if the vertical temperature gradient is small, the residual may not be associated with vertical heat diffusivity. Therefore, we require a vertical temperature gradient greater than $0.0003~^\circ\mathrm{C}~\mathrm{m}^{-1}$ within 5 m below the MLD for the residual to be considered representative of vertical diffusion. However, as noted by Cronin et al. (2015), in the presence of strong currents, the residual term is primarily influenced by heat flux convergence in a stratified shear flow.

To minimize this effect, we impose an additional filtering criterion where the residual is considered representative of vertical diffusion only when horizontal advection remains within two standard deviations of its mean value.

The formation of a barrier layer (BL) inhibits heat mixing below the ML. When the BL is thin (less than approximately 15 to 20 m), the MLD and the isothermal layer (THERM) are close to one another, typically resulting in a negative temperature gradient at the ML base. In contrast, when the BL is thick, the base of THERM is significantly deeper than the MLD, leading

to a more uniform temperature profile within the BL, which can result in small temperature gradients or even temperature inversions. The barrier layer thickness (BLT) is defined as the difference between THERM and MLD when THERM is deeper than MLD (Sprintall and Tomczak, 1992). Here, we define THERM as the deepest level at which temperature decreases by at least $0.2^\circ\mathrm{C}$ relative to the 10 m-depth temperature, following Gévaudan et al. (2021). To ensure that the residual reflects vertical diffusion, we impose a final criterion: the BL must either be thinner than 15 m or contain a temperature inversion of

magnitude greater than $0.2^\circ\mathrm{C}$.

In summary:

$$\text{Residual represents vertical diffusion where} \begin{cases} \frac{\mathrm{dT}}{\mathrm{dz}} > 0.0003~^\circ\mathrm{C}~\mathrm{m}^{-1}~5~\mathrm{m~below~the~MLD~or} \\[2ex] \nabla_\mathrm{h} \cdot (\overrightarrow{u_\mathrm{h}} \cdot \mathrm{T}) < 2 \cdot \sigma_{\nabla_\mathrm{h} \cdot (\overrightarrow{u_\mathrm{h}} \cdot \mathrm{T})}~\mathrm{at~the~MLD~or} \\[2ex] \mathrm{BLT} < 15~\mathrm{m~or} \\[2ex] \mathrm{BLT} > 15~\mathrm{m~and}~\Delta\mathrm{T} > +~0.2~^\circ\mathrm{C~within~the~BL}. \end{cases} \quad (\text{B1})$$

In the second condition, the subscript h indicates that only horizontal derivatives are considered.





## Appendix C:  Coupling Coefficient Linear Regressions

This section presents the binned linear regressions between mesoscale anomalies whose slopes are displayed in Fig. 3: the coupling coefficients. A discussion follows in Subsection 4.2.

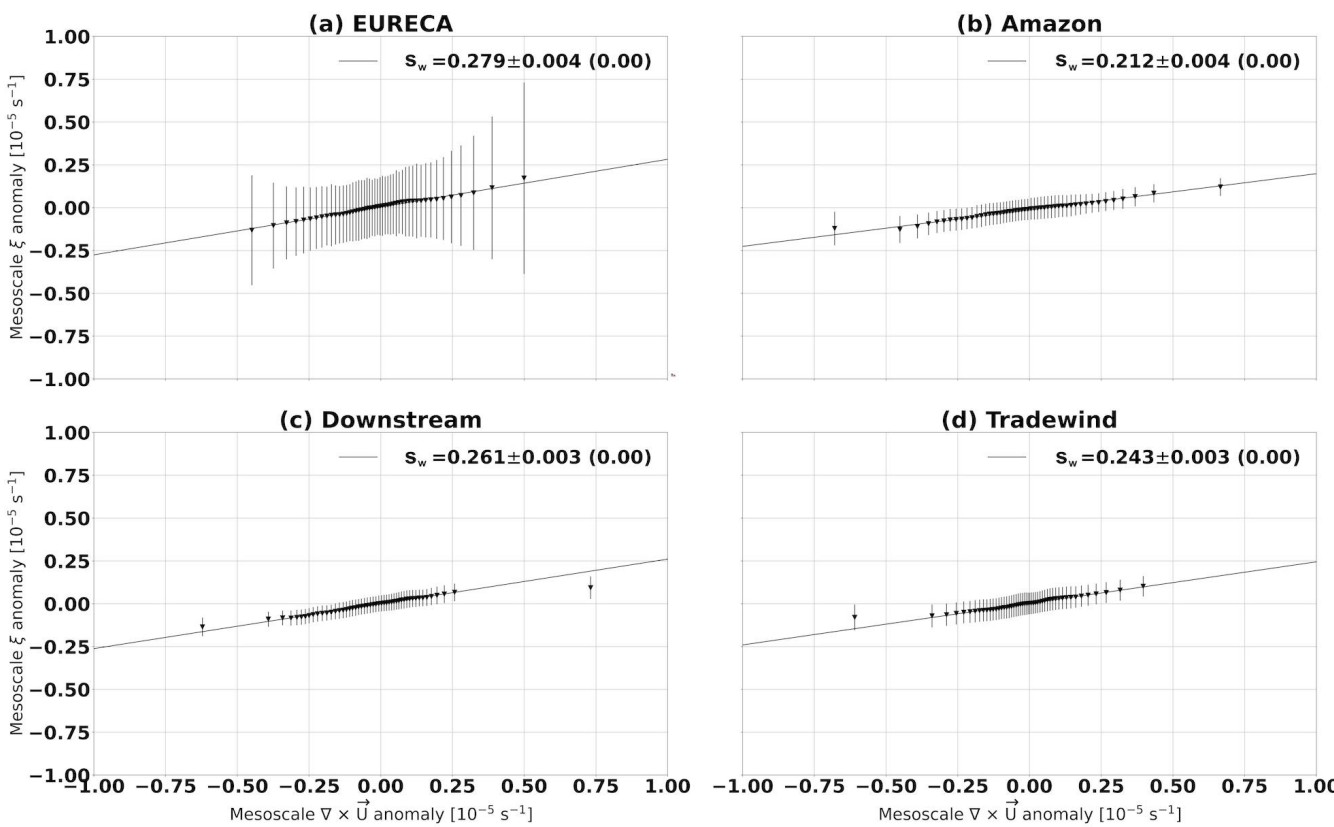

**Figure C1.** Binned scatter plots of surface current vorticity ($\xi$) versus surface wind curl mesoscale anomalies in the (a) EURECA, (b) Amazon, (c) Downstream, and (d) Tradewind. In all cases, error bars represent the standard deviation of each bin. Bins are computed using daily averages from the JF 2020 season, excluding the continental shelf (seafloor depth < 100 m) and islands as described in the main text. All the panels include least-squares regression lines for the mesoscale and submesoscale, with the slope ± standard error (p-value) indicated in the legends. To ensure a sufficiently large number of points per bin, surface current vorticity samples are divided into 2% percentile intervals, each containing at least 15 000 values.

## Appendix D:  Downward Momentum Mixing or Pressure Adjustment?

As mentioned in the main text, ocean mesoscale structures control the overlying surface winds through two mechanisms: Downward Momentum Mixing (DMM) (Hayes et al., 1989; Wallace et al., 1989) and Pressure Adjustment (PA) (Lindzen and





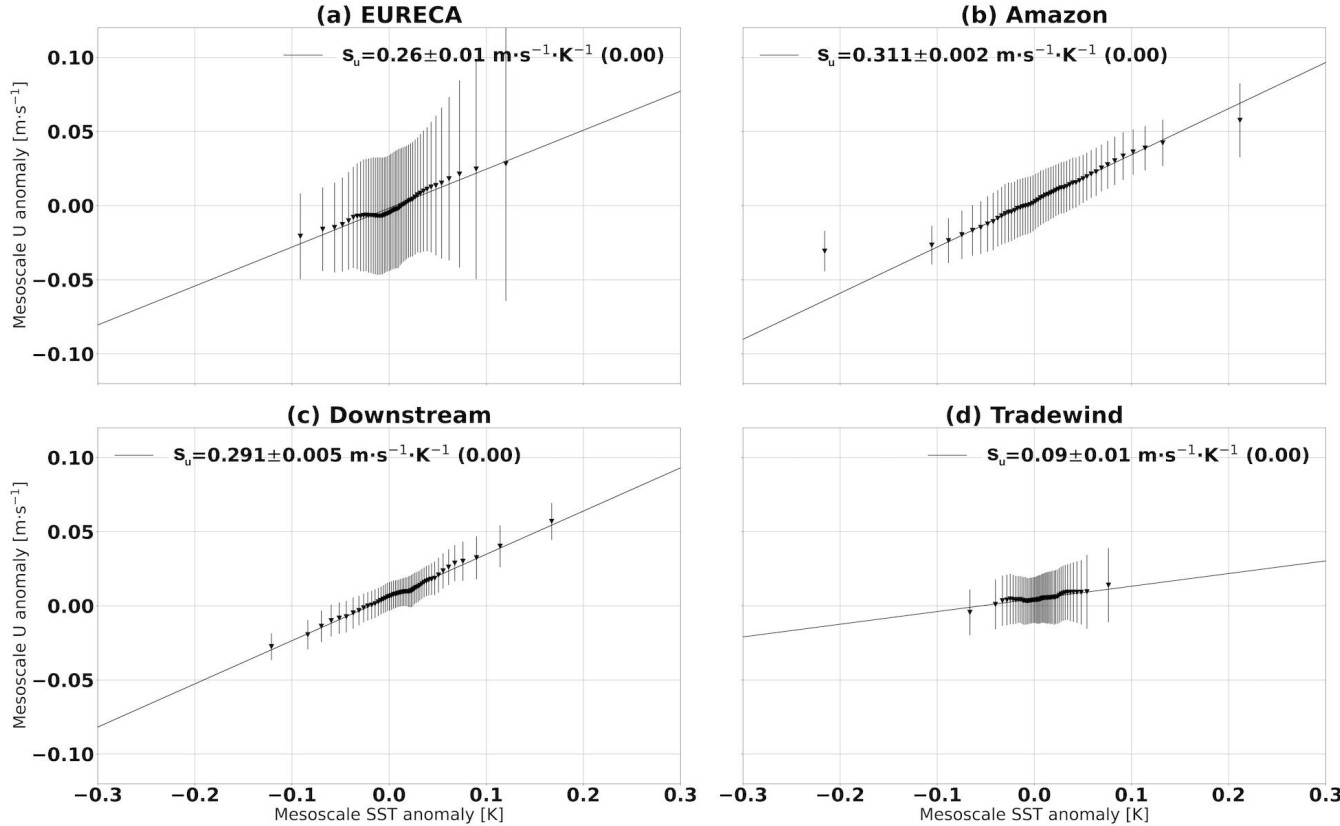

**Figure C2.** As in Fig. C1 but for the linear regression between near-surface wind (U) and SST mesoscale anomalies.

Nigam, 1987). Following the metrics introduced in (Meroni et al., 2022), we test in this section which mechanism (DMM or PA) is taking place more predominantly in EURECA, Amazon, Downstream and Tradewind so as to better understand the SST-U mesoscale coupling. If the PA mechanism is active, positive correlations between surface wind divergence and the SST Laplacian field, both computed in the direction orthogonal to the large-scale wind are expected. In turn, the presence of DMM implies a positive correlation between the wind divergence along the large-scale wind direction and the along-wind SST

gradient, computed as the scalar product of the SST gradient and the large-scale wind vector.

Fig. D1 shows the binned scatter plot of the along-wind wind divergence against the along-wind SST gradient mesoscale anomalies for EURECA (Fig. D1a), Amazon (Fig. D1b), Downstream (Fig. D1c) and Tradewind (Fig. D1d). In all the cases, the along-wind wind divergence increases with the along-wind wind SST gradient mesoscale anomalies at a rate of 0.5 m s$^{-1}$ K$^{-1}$ in EURECA which peaks at 0.55 m s$^{-1}$ K$^{-1}$ in Tradewind and decreases to 0.32 m s$^{-1}$ K$^{-1}$ in Amazon. All the slope values

are statistically significant at the 99% level after a two-sided $t$-test as indicated by the p-values in parenthesis. Thus DMM is significantly acting in the modulation of the surface wind field in all the EURECA and the three sub-regions.





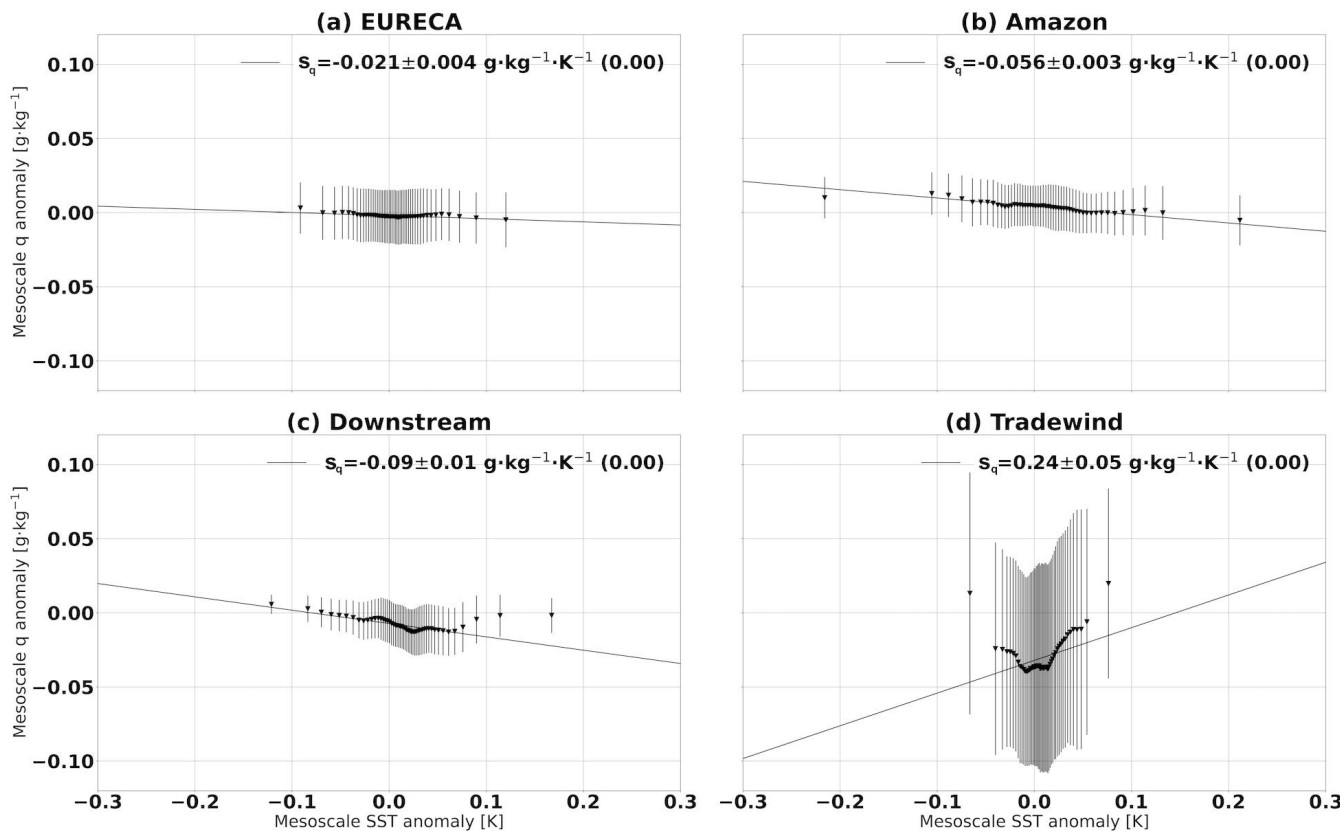

**Figure C3.** As in Figs. C1 and C2 but for the linear regression between near-surface specific humidity (q) and SST mesoscale anomalies.

On the other hand, Fig m D2 displays the binned scatter plot of the across-wind wind divergence against the across-wind SST Laplacian mesoscale anomalies for EURECA (Fig. D1a), Amazon (Fig. D1b), Downstream (Fig. D1c) and Tradewind (Fig. D1d). In EURECA, the slope is negative (and the p-value exceeds 0.01) suggesting that PA is not significant when considering the EURECA region as a whole. The same result was obtained by (Fernández et al., 2023) using satellite observations. The same applies to Tradewind. However, in the warm eddy corridor (Amazon and Downstream), the slopes are positive. This means that PA is acting together with DMM in driving surface wind mesoscale variations. This is, perhaps one of the reasons why we observe such weak wind speed anomalies in Fig. 6b (together with the weak SST anomalies when compared to satellite observations (Fernández et al., 2023) or SST-forced atmospheric simulations (Borgnino et al., 2025) in the same region). However, DMM seems to dominate, as the wind speed vertical and stratification mesoscale anomaly vertical structures are consistent with it.





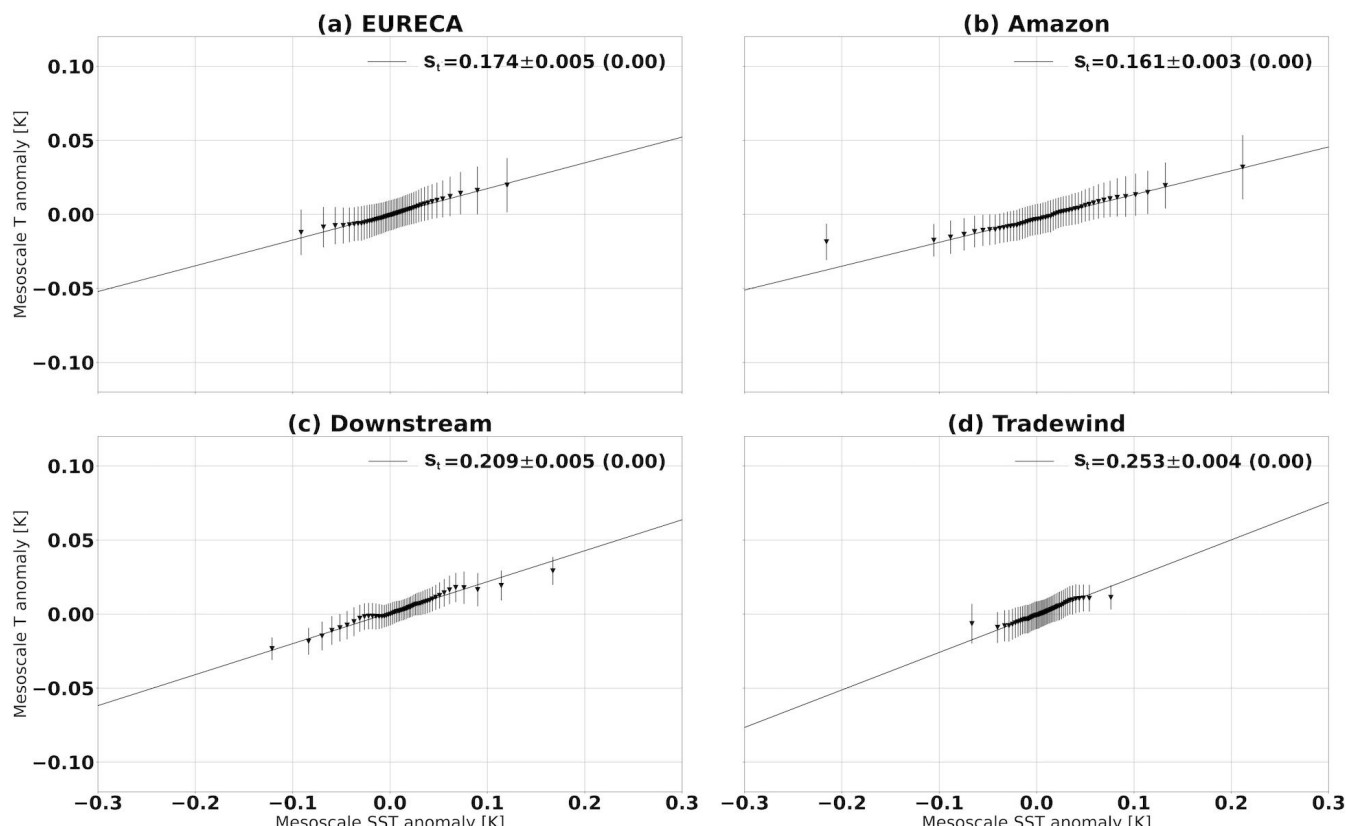

**Figure C4.** As in Figs. C1, C2 and C3 but for the linear regression between near-surface temperature (T) and SST mesoscale anomalies.





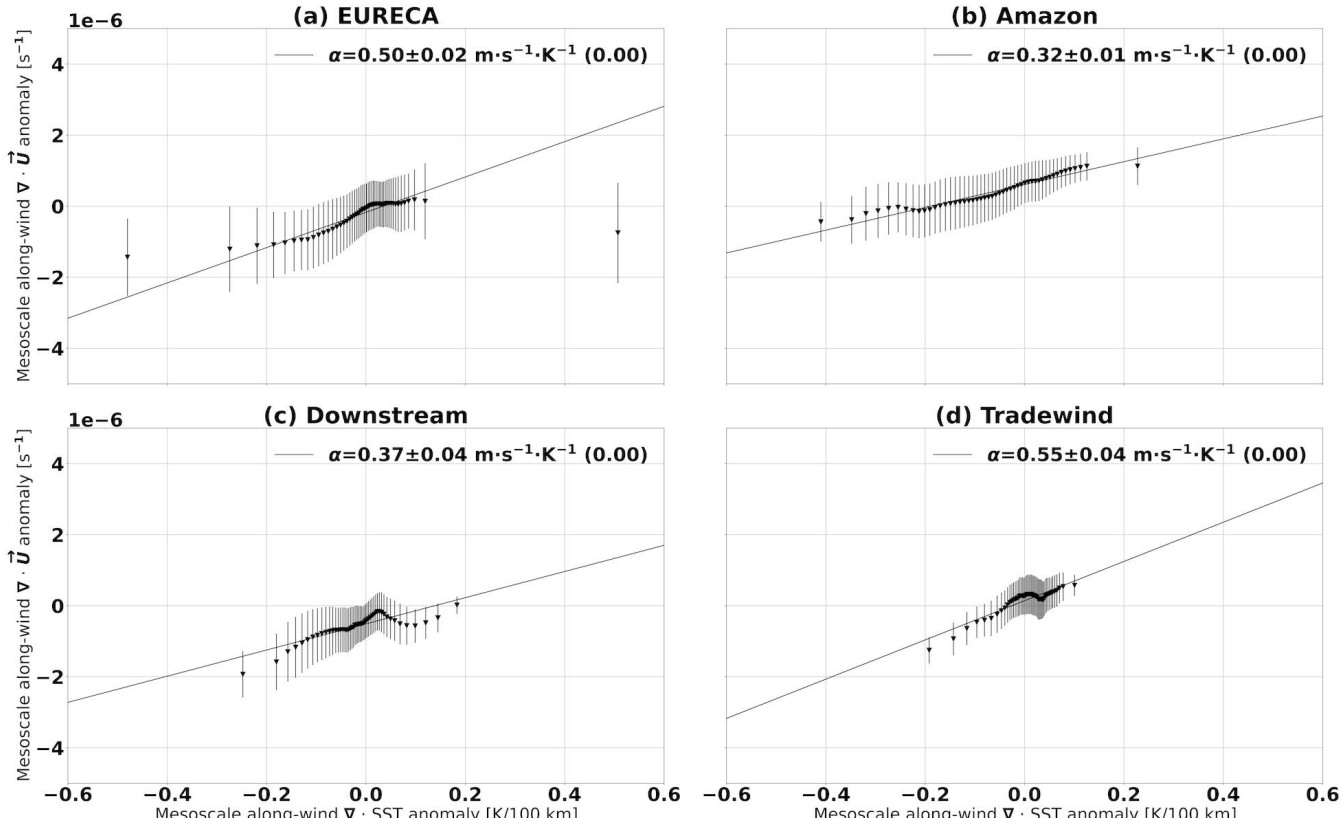

**Figure D1.** Binned scatter plots of the along-wind SST gradient ($\xi$) versus the along-wind wind divergence mesoscale anomalies in the (a) EURECA, (b) Amazon, (c) Downstream, and (d) Tradewind. In all cases, error bars represent the standard deviation of each bin. Bins are computed using daily averages from the JF 2020 season, excluding the continental shelf (seafloor depth $< 100$ m) and islands as described in the main text. All the panels include least-squares regression lines for the mesoscale and submesoscale, with the slope $\pm$ standard error (p-value) indicated in the legends. To ensure a sufficiently large number of points per bin, surface current vorticity samples are divided into 2% percentile intervals, each containing at least 15 000 values.



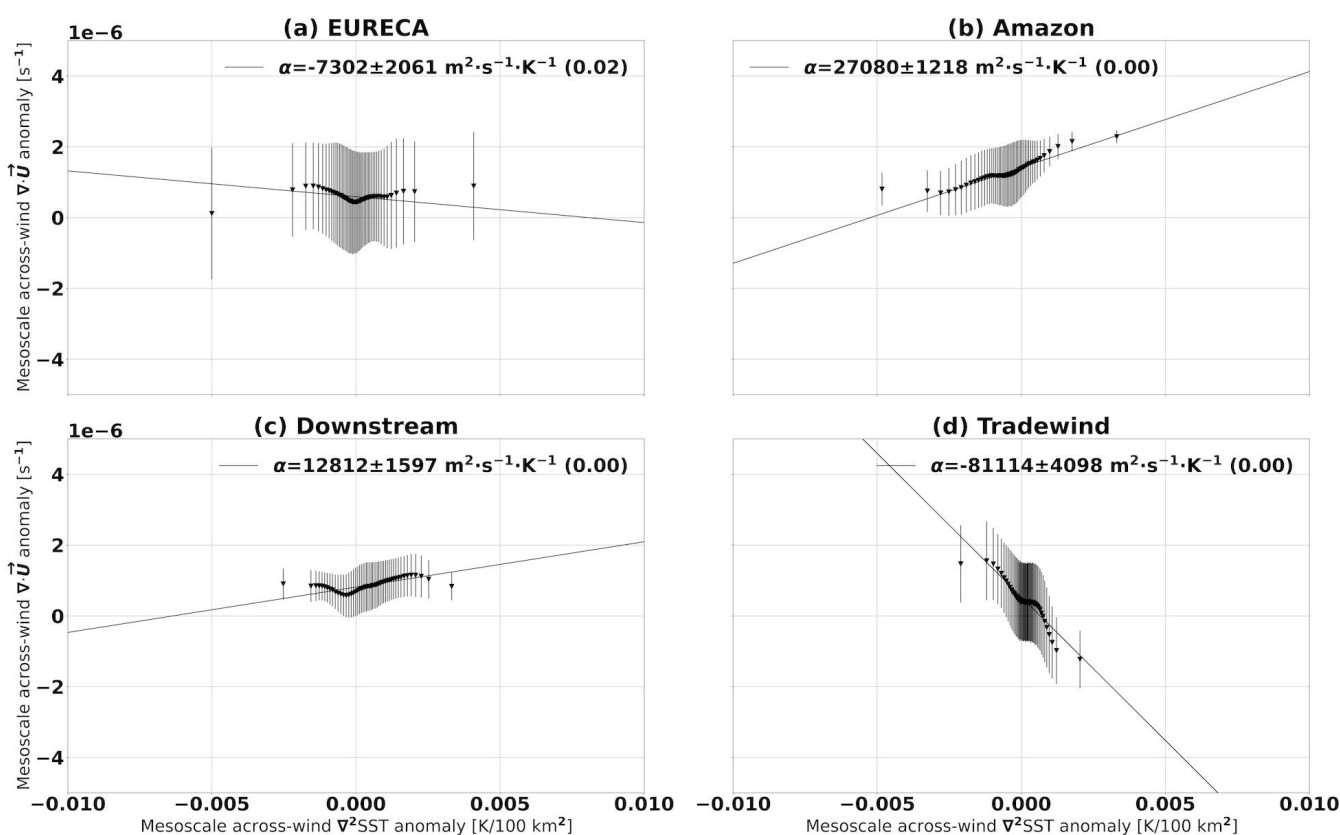

**Figure D2.** Like in Fig. D1 but for the across-wind SST Laplacian versus the across-wind wind divergence mesoscale anomalies.



*Author contributions.* All authors contributed to the conception and design of the study. CC and LR conducted the EURECA simulation, while PF and SS performed the analyses using its output. These analyses benefited from the contributions of CC, LR, FD, CP and GL. GL provided technical support with the computing tools required for processing the model data as well. PF drafted the initial version of the
manuscript, and all authors approved the final submitted version.

*Competing interests.* The authors declare that the research was conducted without any commercial or financial relationships that could be construed as a potential conflict of interest.

*Acknowledgements.* PF was supported by a Ph.D grant from Sorbonne University. This research has been supported by the European Union's Horizon 2020 research and innovation program under grant agreements no. 817578 (TRIATLAS), the Centre National d'Etudes Spatiales
through the TOEddies and EUREC4A-OA projects, the French national program LEFE INSU, the IFREMER, the French vessel research fleet, the French research infrastructures AERIS and ODATIS, IPSL, the Chaire Chanel program of the Geosciences Department at ENS, and the EUREC4A-OA JPI Ocean and Climate program.



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
