# Peer review of "On the Mechanisms Driving Latent Heat Flux Variations in the Northwest Tropical Atlantic: a Modeling Approach"

_EGUsphere, 2025_

## Referee Comment (RC1)

Review of "On the mechanisms driving latent heat flux variations in the Northwest Tropical Atlantic: a modelling approach." By Fernandéz, Speich, Conejero, Renault, Desboilles, Pasquero and Lapeyre, submitted to Ocean Science/EGUsphere, 2025.

**Summary**

The paper aims to investigate the processes causing surface latent heat flux (LHF) variability in the Northwest Tropical Atlantic offshore of Brazil and the Amazon. Selected sub-regions are chosen – one close to the Amazon, one downstream and one in the open ocean "Tradewind". A very-high resolution (1-2km) coupled regional model is used (as in Conejaro et al.). The role of SST, ocean current, and atmospheric boundary layer in affecting LHF is examined. A "downscaling" approach is used, based on filtering and coupling coefficients. The paper finds an important role for atmospheric humidity variations in causing much of the LHF variability.

Overall, I find the paper interesting and appropriate for the journal, and covers the important topic of drivers of latent heat flux variations: this flux is important to both the atmosphere (buoyancy changes, convection etc.) and the ocean (affecting both temperature and salinity via evaporation, etc.) The choice of model seems good, and the diagnostics are impressive. I believe it is suitable to publication after the major and minor points below are addressed.

**Major points**

- 1. Role of humidity. I found myself confused by the role of humidity. The confusion may be due to my ignorance, but I think the paper could try and explain the role of humidity in a clearer fashion. The paper concludes that atmospheric humidity is the main driver of LHF variations, which is supported by Fig. 4 etc.. However some other figures present results which confuse me. Fig. 6d shows extremely weak specific humidity variations near the surface, compared to larger saturation specific humidities (Fig. 6c). I believe the saturation specific humidity anomalies are due to air temperature anomalies strongly controlled by SST. The coupling coefficients also show a weak response of specific humidity to SST (lines 282-292).
  - Despite these weak variations in specific humidity, you seem to show it is enough to be the main cause of LHF variations. I find this part confusing. I would appreciate the role of humidity to be better explained, particularly the connection between Fig. 6 and Fig. 4.
- 2. Novelty. This is the third paper led by the first author on this topic (the title of the paper is almost identical to that of the earlier JGR paper). I understand that this is OK people

can build careers on the same topic – and that the topic is important. However I would like to see the distinction between this and the other papers made more clear. This new paper uses an ultra-high resolution coupled model, which is great, but why do you expect this to give novel results compared to the other papers? Perhaps in the introduction you can state any limitations that arose in the previous papers (using reanalysis, satellite, in-situ data) and describe why the ultra-high simulation can fix some of these limitations and provide new results.

**Minor Points**

Introduction. Add some motivation on why we are interested in latent heat flux for the atmosphere – possible clouds, convection etc.

Line 171. At this point, give some background as to why you are using the downscaling method, and why not just use the high-pass spatial field.

Fig. 1b is a bit complicated, especially for the first figure of the paper. I suggest to move the details of the relative wind and CFB to section 3.2 (or create a new section 3.2) and move Fig. 1b to a new Fig. 2

Caption of Fig. 4 and other locations. At a few points in the manuscript, the large number of samples (e.g. 15,000) is mentioned. It is worth noting that for statistical significance testing, one should use the number of independent samples. (For example, many samples may be over the same eddy.) There are ways to take this into account, e.g. using the autocorrelation of the data.

Lines 543-545. This needs more explanation. Does adding the downscale method actually enhance representation of air-sea fluxes in the model, or is it just a diagnostic?

Appendix D. This is interesting but it feels a bit out-of-place. Where is Appendix D referenced in the manuscript? I feel like it could be expanded upon and presented elsewhere, as it is not central to the paper.

**Very minor + Grammar/wording**

Lines 6-9. Please write without the parentheses for different cases.

Line 28. (THFs, comprised of latent and sensible heat fluxes).

Line 30 which include-> the latter including

Line 33. 250km seems too precise-> "about 250km"? But, it probably varies regionally.

Line 33. Typo on Gill 1982

Line 39. Add "and coupled models (Small et al. 2019)"

On this theme, there is a body of literature on stochastic coupled and ocean-only models which are relevant here (Frankignoul and Hasselman1977, Barsugli and Battisti 1998, Frankignoul et al 1998, Wu et al. 2006, Bshop et al. 2017, Laurindo et al. 2022, JGR). You do not have to reference them here, but they should be of interest. Most are cited in Bishop et al. (2017).

Line 50 "from hours to weeks to long-term climatologies (e.g. Chelton et al., Minobe et al. 2008). (I am personally interested in how the processes change between hours and days and months – I may be giving my name away by referencing here Small et al. 2023, J. Clim..)

Lines 280-281 and 314-318 (and to a lesser extent 416-422). I think that comparison with work done by co-authors using the same simulation may be considered just a "sanity-check" or consistency-check. It is an important testing procedure but probably should not be cited. It would be better to compare with an independent study if available. However it is fine to compare your model analysis with your work on in-situ and reanalysis data (which you do later) as these are independent datasets.

Line 52. "mesoscale eddies" -> "and fronts"? Some of the referenced papers discuss time-averaged fronts (e.g. Minobe et al. 2008).

Line 57. "resolution is increased" - > "grid spacing is reduced"

Line 62. "currents and winds are aligned, and surface stress reduced"

Lines 64-65. "currents oppose surface winds, and surface stress is increased".

Lines 97-98. Reword without parentheses for clarity.

Line 168. Wording is a bit confusing. Suggest "same variables as LHF\_u, but smoothed with..."

Line 229. "As in Gevaudan..."

Fig. 2. Can you add bathymetry onto one of these panels?

Lines 249-250. Some mis-spelling and grammar issues.

Line 258. Replace with "lacks strong temperature and salinity signatures at the surface."

Sentence beginning line 272 could be moved down to the discussion of s\_u (line 276).

Lines 304-305. I think smoothed variables are not obtained by subtracting low-pass values from the original data.

Fig. 5 labelling is hard to follow. Suggest to use words on the horizontal axis labels, e.g. "Relative minus absolute wind" (blue) and "Relative wind (no CFB) minus Relative wind (CFB)" (orange) and use the corresponding equations in the text.

Line 369 "structure in Amazon region"

Line 379. Give equation for N^2.

Line 383. The pressure adjustment can also give dipoles due to a secondary circulation – Small et al. (2003 (Tropical Instability Waves)).

Section 4.4 could be separated into a sub-section on atmosphere and one on the ocean (with the air-sea interface discussed between the two perhaps).

Fig. 8. Black contours in panels c-h are salinity?

Line 492-493. "fully resolving ocean mesoscale" is a motivation that could be put in the Introduction – see major comment on Novelty.

Line 508. Wording issue, should be "negative in Amazon and downstream".

---

## Author Comment (AC1)

**REPLIES TO REVIEWER 2**

We thank the referee for the careful reading of the manuscript and the constructive comments. We have thoroughly considered all remarks and provide detailed responses below. We believe that the suggested revisions have significantly improved the overall quality of the manuscript. The answers to the points raised by the referee are presented in the following sections.

**Major comments**

**Line 138: What is meant by "Amazon sub-region is influenced by the Amazon plume". Do you mean that there is advection of warm, fresh surface waters into the sub-region? Please be more specific as the influence the authors have in mind ends up being important throughout the text. Without stating this influence up front here, some later discussions are confusing.**

We warmly thank the reviewer for this remark. In the revised manuscript, we now clearly state that warm, fresh surface waters are advected into the Amazon sub-region and that this advection is associated with the Amazon freshwater plume. The corresponding modifications can be found between lines 144 and 145 of the revised manuscript.

**Lines 203-205: This sentence is unclear. It sounds like you will look into linkages between mesoscale SST anomalies and the Amazon plume, but the Amazon plume is not present in your EURECA box (per line 244). Also, as worded, it sounds like you are saying that the mesoscale SST anomalies lead to the Amazon plume, which does not seem to make sense.**

We thank the reviewer for this remark. Negative SSS anomalies associated with the Amazon plume are indeed usually warmer, as Amazon river waters are typically warmer than surrounding ocean waters. The reviewer is correct that this relationship is not clearly visible in the climatology fields shown in Fig. 3. This is due to the high variability of the ocean surface circulation in this region: the warm and fresh Amazon plume interacts with the cyclonic North Brazil Current Ring C1 (see figure below),which advects cooler waters from the cold filament. When averaged over two months, this variability obscures the typical correspondence between low SSS and high SST. The  figure below of snapshots of SST (left) and SSS (right) illustrate this variability. To avoid potential misinterpretation, we have removed from the main text all statements inferring a warm–fresh relationship directly  from the climatologies. In addition, we have reformulated the oceanic analyses of section 4.4 where this relationship is assessed to further elucidate the linkages between the Amazon freshwater plume and SST mesoscale anomalies.

In the lines the reviewer refers to, our intention was to highlight that the positive SST anomalies associated with the Amazon freshwater plume (which do not appear clearly in the climatology for the reason above) are nonetheless part of the ocean mesoscale: plume waters exhibit a positive mesoscale SST anomaly. This establishes a clear connection between a well-known physical feature (the Amazon plume) and the broader concept of mesoscale SST anomalies, which is central to our analysis. The purpose of computing the mixed-layer heat budget is precisely to diagnose the processes of heat redistribution

associated with the Amazon plume that contribute to the maintenance or dissipation of surface mesoscale SST anomalies.

We have also removed the statement in line 244 of the previous version claiming  that warm SST anomalies east of the Amazon subdomain are associated with the northward advection of warm Amazon plume waters, as the snapshots show that this interpretation is not supported.

Please, find the rephrasing of lines 203-205 of the old version between lines 214 and 218 of the new one. There, we include as well a definition of the Amazon plume waters based on observational studies in the region (SSS<35psu, Reverdin et al., 2021, https://doi.org/10.1029/2020JC016981)

P.S. We may be misunderstanding part of the reviewer's question, but the Amazon plume (now defined in the main text as waters with less than 35 psu) clearly enters both the EURECA region (see Fig. 2 in the revised manuscript) and the Amazon subregion (see Fig. 3d, which shows the JF 2020 model climatologies).

[Figure]

**Lines 241-242: Can the authors provide more discussion on the cold filament of surface water across your domain? This is a very prominent feature of your experiment region and should be discussed further as background for the remaining analyses.**

We thank the reviewer for this comment. This is indeed a very important feature, with notable implications for air-sea interactions, as discussed below. The cold filament is generated by the advection of deeper, colder waters toward the surface, as illustrated in the transect shown below (8°N, 9th February 2020). We have included two sentences in the main text to clarify this point. These additions appear between lines 251 and 255 of the revised manuscript.

[Figure]

**Lines 255-256: This seems to contradict what is said above, as the low salinity patch extends within the Amazon box while the warm SSTs do not, but both are said to be related to the Amazon plume.**

The reviewer is correct, and we thank them for this remark. As noted in our responses to earlier comments, the temperature signature of the Amazon plume cannot be reliably

inferred from the SST climatology. The mean SST field is highly variable because the warm plume interacts continuously-with the cold filament, and this variability obscures the underlying relationship. When examining instantaneous fields (see Fig. 8 of the revised manuscript), lower SST values are generally associated with higher SSTs; however, this correspondence does not hold uniformly through the plume. We now make this point explicit in the text (see lines 268-269 of the revised manuscript).

We also note that Fig. 2 in the previous version (model climatologies) contained an error: the climatology was computed using December-January-February 2020 rather than January-February 2020 alone-. This has been corrected in the revised Fig. 3 (previously Fig. 2), which now more clearly shows the low-SSS/high SST relationship.

Finally, we have removed the reference to the Amazon plume at this point in the manuscript and now address it explicitly in section 4.4.

**Line 372: Again, it is indicated that the Amazon plume is present in the EURECA domain, contrary to what is said on line 244. Discussion around the Amazon plume need clarifying throughout the text, as what remains of the plume in your study area and its impacts on the Amazon box are unclear and inconsistent.**

We thank the reviewer for this remark. The Amazon plume is indeed present in the Amazon subregion, as illustrated in Fig. 3d (showing the low SSS patch < 35 psu crossing the Amazon subregion) and Fig.3a (which displays high SSTs over the low -SSS patch, though, as noted in earlier responses, the SST signal is noisier due to interactions with the cold filament located south and west of the Amazon subregion). We hope that the updated Fig. 3 improves the clarity of this discussion. In addition, to avoid misinterpretation, we have removed all references to the Amazon plume from Section 4.1 (including the previous mention line 244) and now restrict the description in that section to the features directly observable in the climatologies.

A detailed analysis of the Amazon plume is now provided in Section 4.4. In this section, we apply the definition of plume that we have added the methodology (SSS < 35psu; see line 218 of the revised manuscript).

Finally, we have reformulated the sentence previously located at line 372; the updated version can be found within the paragraph spanning lines 390–396 of the revised manuscript.

**Lines 356-357: Wind variations in Fig. 2c are very difficult to see. Are the contributions to the relative wind variability mainly due to the differences in the surface currents in the three regions or is the variability in the winds higher in Amazon and Tradewind boxes? Showing time series of the winds and currents and/or the relative wind for the four regions might be more helpful than comparing their separate time means for the purpose of this discussion.**

We warmly thank the reviewer for this remark. The figure below represents the time series of the area mean surface wind speed (blue), relative wind speed (black) and surface current

(red). We can see that the variability of the winds is higher in the Tradewind box, which experiences variations between 4 and 12 m/s than in Downstream or Amazon. We can also appreciate that relative winds are closer to surface winds in the Amazon and Tradewind boxes than in the Downstream box. This explains why we observe LHF differences (blue markers) in the positive and negative sides of the x axis in Figs.6b and d of the new version of the paper. However, in the Downstream region surface currents are stronger and generally aligned with surface winds (see Fig. 3e of the new version of the paper). Therefore, relative winds are weaker than surface winds throughout the two months.

We have included this figure in the supplementary as we think there are already enough figures in the main text. However, we cite it in the discussion The corresponding modifications can be found between lines 366 and 383 of the revised version.

[Figure]

**Lines 365-368: If CFB always increases surface winds in the direction of the current, then this particular process would result in relative winds that are smaller than the wind alone, correct? So why would this effect ever increase LHF? It seems like this discussion conflates the relative versus full surface wind impacts on LHF with the impacts of CFB alone. Or maybe I am not understanding CFB?**

We thank the reviewer for this remark. Please, let us clarify this point with an example and let us refer to Fig. 1b of the new version of the manuscript. In particular, let's focus on the left hand side of the eddy, where surface currents (black arrow) and winds, which include CFB (blue arrow) are aligned.

Taking into account CFB, the relative wind is just the difference between the blue and the black arrows, shown as a green arrow in the schematic. However, within the surface wind's blue arrow, there is the momentum imprinted by surface currents (represented by the orange arrow, this is, the CFB effect). If we want to obtain a surface wind vector without the effect of CFB, we must perform the operation blue arrow minus orange arrow. This is equivalent to removing the CFB-induced wind speed from the total surface winds. Since surface winds and currents are aligned, this operation results in a vector smaller in magnitude than the original surface winds (blue arrow).

If we now compute the relative winds between this surface wind without the CFB effect and surface currents, we obtain a smaller value than if we had computed relative winds with the original surface wind which accounts for CFB. Since LHF is proportional to relative winds, the LHF computed without the effect of CFB will be smaller than the one obtained if CFB is accounted for. Therefore, CFB increases LHF in this case. We hope it is clearer now.

We have included a more detailed explanation of this schematic (Fig. 1b of the new version) between lines 62 and 69 of the new version.

**Line 385: Is not that the wind speed increases aloft and decreases at the surface over cold SSTs, but rather that the momentum transfer from aloft to the surface just does not occur. This is the "decoupling" of the surface layer from the free troposphere common in stable boundary layer situations. The wording here is misleading as it implies an opposite momentum transfer to what is happening over warm SSTs.**

We thank the reviewer for this remark. We agree that the momentum transfer from aloft to the surface does not occur over cold SSTs. However, the anomaly histogram referenced by the reviewer (Fig. 7b in the revised version) indeed shows a reduction of near-surface winds (negative anomalies) and an enhancement aloft (positive anomalies). This vertical "decoupling" inhibits downward momentum transfer: surface winds weaken, while winds aloft strengthen because their momentum is not extracted. We have now explicitly stated that momentum transfer is suppressed in this context, as pointed out by the reviewer. The corresponding revision appears in lines 409-410 of the revised manuscript.

**Line 387: Why show saturation specific humidity rather than potential temperature to illustrate temperature differences?**

We thank the reviewer for this comment. We agree that potential temperature can be useful for assessing vertical stratification. However, we chose not to include it here since we already analyse the distribution of the Brunt-Väisälä frequency in Fig. 7a. To clarify the relationship, we have added the equation for $N^2$, as suggested by the other reviewer (see line 399 in the revised manuscript). Instead, we present the saturation of specific humidity since it is required to compute the specific humidity deficit, a key factor in the LHF bulk formula. In our analysis, variations in saturation specific humidity dominate over those in the

specific humidity (Fig. 7d) in controlling changes in specific humidity deficit (Fig. 7e), thereby modulating the total LHF: higher LHF values occur where the surface saturation specific humidity is larger.

While we acknowledge that within the saturation specific humidity formula there is a dependence on air pressure (which can significantly vary in the first 2000m), the saturation specific humidity is useful when assessing air temperature variations at a given height as a function of the SST mesoscale anomaly. Finally, from a practical point of view, we only need it at the surface to compute LHF, so in theory we could suppress panel 7c. However, for the sake of completeness we would like to keep it.

**Lines 407-408: On line 355 you state the opposite, that the LHF variations due to SST mesoscale variations are mainly due to the dynamic contribution. I believe this confusion is due to the many ways these authors use the term "dynamic contribution". On the one hand, it seems to be used to describe part of the overall "thermodynamic contribution" as on line 355, while on the other hand it is also used to describe the relative wind impacts which seem to be what is meant here? I think the terminology needs to be consistent throughout the text given how many effects are being examined. It is difficult as a reader to keep them all straight. And Fig. 1 only shows two of them. Perhaps, the authors can provide a table of effects they are investigating along with a description of what they are and how they are isolated? In any case, for the discussion on lines 407-408 can the authors return to their LHF naming convention and add the appropriate terms in parentheses after the words "thermodynamic contribution" or "SST changes" "variations in specific humidity" and "wind speed" or "dynamic contribution" so we know which term to refer to in Figs. 4 and 5.**

We thank the reviewer for pointing out this inconsistency in terminology. As stated in line 335 of the previous version of the manuscript, the dynamic contribution indeed dominates the total LHF variations. In that line, the dynamic contribution was defined as "the LHF variations linked to the SST-induced modification of the near-surface atmospheric variables". For clarity, we now explicitly specify that the thermodynamic contribution corresponds to the component of LHF changes driven solely by SST-induced variations in saturation specific humidity, whereas the dynamic contribution encompasses the LHF changes arising from SST-driven modifications of the near-surface wind speed and specific humidity. We have revised line 335 accordingly and ensured that this terminology is used consistently throughout the manuscript (see line 347-351 in the revised version).

Concerning the discussion in lines 407 and 408, it has been rephrased so that it aligns with the discussion between lines 347-351 of the new version (335 of the old one). Please, find the modifications between lines 442 and 448 of the revised version of the paper.

We would also like to clarify. that Fig. 1 is not intended to represent the "thermodynamic" or "dynamic" contributions. Instead, it illustrates the downward momentum mechanism (Fig. 1a) and the current feedback (Fig. 1b).

Following the reviewer suggestion, we have added Table 2, which summarises the physical processes considered and the LHF differences used to isolate each of them.

**Line 425-429 - This entire discussion is difficult to follow. The authors refer to the distribution of the mesoscale SST anomalies in space but we only see a histogram with height in Fig. 6. Also, the Amazon plume is mentioned multiple times despite it not being within the EURECA domain. Please clarify what is meant by the "core of the Amazon plume" since Fig. 1c suggests the plume is mostly outside of this domain.**

We thank the reviewer for this remark. The reviewer is right that the distribution of mesoscale SST anomalies in space cannot be extracted from a histogram. Therefore, we have reformulated all the analyses following the atmospheric vertical profile histograms and before the mixed layer heat budget so that the conclusions are more accurate. We now present snapshots of SST and currents (Fig. 8) and clearly mark the boundary of the Amazon plume with the 35 psu isoline as suggested by previous observational research (Reverdin et al., 2021, https://doi.org/10.1029/2020JC016981). According to this definition, the Amazon plume crosses the Amazon sub-region in February 2020.

In addition, we present maps of the mixed layer depth, barrier layer thickness and OSS index integrated down to the mixed layer to discuss the spatial variations we were aiming to point out in the all version.

Please, find the changes in sections 4.4.2 and 4.4.3 of the revised version.

**Lines 446-448 - The colormap may be too hard to read for the OSS values, as it seems that they never get higher than 40% (cyan). A value of 50% would be green, and at least in the version of the figure provided in the manuscript there does not seem to be any green color. Also, is it correct to say that salinity is important to the stability when overall the OSS % is well below 50%? According to Line 235, that means salinity is not important to the ocean stratification. Also, the core of the plume appears to be from the surface to about 20m depth, while the peak OSS values appear to be from about 15 to 40 m depth. So the peak OSS seems to occur at the base of the plume, not in the core of the plume.**

We thank the reviewer for pointing out this feature. The figure the reviewer is referring to is no longer in the manuscript since a spatial analysis cannot be performed with mesoscale SST anomaly - depth histograms. Instead, we have plotted in Fig. 9d of the new version the averaged OSS index in February 2020. We also have changed the palette and plot limits to highlight the differences.

 As the reviewer states, all the OSS values are below 50%, meaning temperature is always the main driver of stratification. There are certain regions (i.e. to the northwest of the plume) where OSS reaches 30%, meaning that the salinity contribution to stratification increases although it is still smaller than the temperature one. All these modifications have been incorporated into the revised manuscript (see lines 510 – 515).

**Lines 440-464 - It might help this discussion (as at lines 424 and 426, for example) to provide an additional panel in Fig. 7 showing the SST across the Amazon box only with the SST mesoscale anomalies overlaid as contour lines. This will help to see spatially where the mesoscale SST anomalies are located within the domain. The dT/dz panel could be removed as it confirms no temperature inversions. This could simply be stated in the text without a figure. Also, or alternatively, the Amazon plume waters could be delineated in at least panel 7d.**

We thank the reviewer for these two comments. As suggested, we have removed the dT/dz panel from Fig. 9 (in fact we removed the whole figure to substitute it by maps). We acknowledge we checked the absence of temperature inversions in lines 487 and 492 of the revised version.

In addition, we now include several SST and surface current snapshots in Amazon, with the plume delimited with the 35 psu isoline in magenta (Fig. 8) and a February 2020 average of SST (shading), salinity (contours), surface currents (arrows) and Amazon plume boundary (magenta contour, the 35 psu isoline) in Fig. 9a, providing a planar view of all these fields. We hope this improves the clarity of the discussion.

**Lines 452-455 - Still confused where the Amazon plume waters are with respect to the mesoscale SST anomalies. If it is represented by the most fresh SSS anomalies (mesoscale SST anomalies of ~-0.02 to 0.2), then the total heat flux over this plume is near zero, not transitioning to negative until mesoscale SST anomalies > 0.2. Heat tendency in the plume is also near zero, with some positive heating at the upper end of the mesoscale SST anomalies within the plume (>0.1). Again, this discussion might be clearer if we had a planar view of the mesoscale SST anomalies within the Amazon sub-region with the Amazon plume clearly delineated on the map either in Fig 6 or Fig. 7.**

We thank the reviewer for pointing this out, and we hope that the new analyses and figures in sections 4.4.2 and 4.4.3 of the revised manuscript clarify this issue. Please refer to Fig. 8 for several snapshots of the SST field, the surface currents and the boundary of the plume (35 psu isole) and to Fig. 9 for an analysis of the vertical structure of the mixed layer depth averaged in February 2020, also with the plume boundary overlaid as a magenta contour.

Following the reviewer's advice, we have removed the 2-d ocean histogram (Fig. 7 of the old version) as it does not allow a spatial analysis of the anomalies).

**Fig. 8 - Please plot SSS over SST in one of these panels so we can see the salinity signature of the Amazon plume and its corresponding SST signature together. This will make earlier discussions of the plume influences easier to follow. You could also consider such a panel for Fig. 2 for the entire EURECA domain.**

We thank the reviewer for this comment. As suggested, we have added the boundary of the Amazon plume (35 psu isoline) in Figs. 8, 9 and 10 of the revised manuscript where we perform the oceanic analysis of Amazon. We have not included it in Fig. 3 of the revised manuscript (model climatology, the old Fig. 2) to reduce the complexity of the figure (which is already charged with shading, arrows and/or contours). However, we have modified it so that it has a discrete palette in the shading fields. This allows to clearly distinguish the SSS isolines in Fig. 3d. We hope these changes facilitate the flow of the discussion.

**Fig.8a vs Fig. 2a - It is not clear that the SST contours in Fig. 8a match the filled contours in Fig. 2a. According to Fig. 2a, the warmest SSTs in the Amazon domain are near the 17.2 contour label for specific humidity and towards the northeast, where colors are more yellow. However, there is a clear tongue of warm SST extending from the southwest across to the northeast of this domain (the Amazon plume) in Fig. 8a. Can the authors use a different color bar in Fig 2a to better highlight the SST gradients across the region?**

We warmly thank the reviewer for pointing out this inconsistency. The reviewer is correct : the original Fig. 2 (now Fig. 3 of the revised manuscript) was based on the DJF mean rather than the JF mean. And the old Fig. 8 only in February 2020.  We have now corrected this and verified that the SST and SSS fields (and the rest of them) shown in Fig. 3 of the revised manuscript are averaged over JF 2020. However, we keep the mixed layer heat budget analysis (now Fig. 10 of the revised manuscript) only for February 2020 since the plume does not arrive to the Amazon sub-region before mid-February 2020.

**Line 473 - Do the authors mean temperature advection from the east? The temperature contours appear to be oriented east-west, with temperature increasing to the west. Advection from the south would bring cold water northward I would think, just looking at the SST contours in panels (a) and (b). Or perhaps there are warmer waters below the surface to the south within the ML?**

We thank the reviewer for this comment. The reviewer is correct: the advection originates from the east. We show this in Fig. 9a of the revised manuscript where we can find the SST field in shading and surface currents in grey arrows. We comment on this in the main text in line 519 of the revised version.

**Lines 474-479 - The temperature tendency within the <35 PSU contour is not just negative, it is positive in the southwestern region of this contour, with heating due mainly to horizontal advection, not atmospheric forcing. The region defined by SST > 26.7 degC and SSS between 35 and 35.4 PSU also seems to not exactly match the very narrow region of positive temperature tendencies. It is not clear where the authors are referring to when they talk about the core of the plume. A panel with SST and SSS together with the plume marked on the figure would facilitate this discussion. Also, this discussion contradicts that on lines 453-454.**

We thank the reviewer for this comment. Below we detail the modifications we have implemented to address it.

Given the heterogeneous structure of Fig. 10a (the total temperature tendency map in the revised manuscript, old Fig. 8c), we have removed the dT/dt panel from the histograms (and in fact, the old Fig. 7 with the ocean histogram). The mean values within each bin were strongly influenced by highly variable dT/dt patterns, which limited the interpretability of the panel. As the reviewer notes, the temperature tendency is not uniformly negative within the interior of the plume (whose boundaries are now defined as the 35 psu isoline as shown in all panels of Fig. 10).

We now clarify that the warmest part of the plume exhibits negative total temperature tendencies, primarily driven by  the negative atmospheric forcing. However, in its southwestern part, the total temperature tendency is positive due to horizontal advection from the east (Figs. 9a and 10b of the revised version of the manuscript). These clarifications have been incorporated along section 4.4.4 (Mixed Layer Heat Budget) of the revised manuscript. They are summarised between lines 539 and 544 of the new version.

**Fig. 8e,f - Panel (e) is not discussed and is an order of magnitude smaller than most of the other terms. Suggest removing this panel. Likewise, although panel (f) is briefly mentioned, this term is also an order of magnitude smaller than the others and could be left out along with discussion on lines 480-482.**

We thank the reviewer for this remark. However, we prefer to retain Fig. 8 (now Fig. 10) with all its panels for the sake of completeness. We have added a brief discussion of the panels that were previously not referenced in the text, so that all panels are now explicitly cited. Please find the discussion between 532 and 536 of the new version.

**Line 487-488 - If this statement were true, would not the temperature tendencies be zero? They are in fact small compared to the advection, residual and atmospheric forcing terms. Is that what the authors are trying to say, despite the discussion on Lines 474-479 describing the tendencies?**

We thank the reviewer for pointing out this inaccuracy. We have removed this sentence since the total temperature tendencies is not zero and it does not add any key information to the conclusions.

**title - Suggest a change to the title as it seems to describe only one section of the manuscript. The latter part of the manuscript is spent understanding the ML budget. Maybe "On the Mechanisms Controlling SST and Ocean Mixed Layer Heat Content in the Northwest Tropical Atlantic: A Modeling Approach".**

We thank the reviewer for the suggestion. As a counterproposal, we suggest the following title:

"Mechanisms Driving Mesoscale Latent Heat Flux Variations and Mixed Layer Heat Content Evaluation in the Northwest Tropical Atlantic"

We believe that latent heat flux should appear explicitly in the title, as it constitutes a central component of our study.

**MINOR EDITS**

**Figure 1 - Suggest splitting this figure into two different figures, one with panels (a) and (b) and one with panel (c). The current 3 panel layout is crowded and the text for panel (b) extends into panel (c).**

We thank the reviewer for this comment. We have implemented the suggested changes, and the two figures in the introduction of the revised manuscript. As suggested by the other reviewer, we have added the seafloor depth as panel b in Fig. 2.

**Line 70-73 - Change "shortens" to "shorten" but also check sentence structure as it does not read well.**

Thank you for pointing this out. We have corrected this mis-spelling and the corrected version of the sentence is found in line 73 of the revised manuscript.

**Line 125 - Do the authors mean freshwater, heat, and momentum fluxes? Turbulent does not make sense in this context since momentum fluxes are also turbulent fluxes.**

We thank the reviewer for this remark. The reviewer is right and the text has been modified as suggested. Please find the corrected version in line 131 of the revised manuscript.

**Line 240 & Fig. 2 - Can the authors add Trinidad and Tobago to these panels?**

We thank the reviewer for this suggestion. A number 1 has been placed over Trinidad and Tobago and a number 2 over a region close to Barbados in Figs. 2 and 3 of the new version (Fig. 3 of the revised version is the old Fig. 2) so that geographical references are easier to follow in the main text.

**Line 249 - Typo, "wuch" should be "such".**

Thank you for pointing out this mis-spelling. Please find the correct spelling in line 262 of the revised manuscript.

**Line 263 - Change to "in the following sections."**

Thank you for the comment. We have changed the sentence as suggested. Please find it in line 276 of the new version.

**Lines 274-275 - Sentence is not grammatically correct. Please fix.**

Thank you for pointing this out. The sentence is corrected and can be found in line 288-289 of the revised manuscript.

**Line 328 - Typo, should be "among" or "amongst".**

Thank you for pointing out this mis-spelling. It has been corrected and can be found in line 340 of the revised manuscript.

**Sec. 4.4 heading - should be "the Amazon"**

The reviewer is correct, thank you for noting this. The title of the section has been modified accordingly (see line 390 of the updated manuscript). In addition, following another reviewer's suggestion and to improve the flow of the discussion, we have divided subsection 4.4. (Vertical structures) into four subsections addressing the atmosphere, the air-sea interface, the ocean mixed layer structure and the mixed-layer heat budget.

**Fig. 6 caption - Please add that the values shown are for the Amazon box only for clarity.**

Thanks for your comment. We have added this information to the caption of Fig. 7 of the revised manuscript (Fig. 6 in the old version).

**Fig. 7 - The labeling on these panels is overall confusing since the x-axis for all panels is only labeled in panels (g) and (h), but a color bar is shown beneath all the panels. It would be better to include the Mesoscale SST anomaly tick labels and axis label in all panels for readability.**

We thank the reviewer for this remark. The figure the reviewer is mentioning in this comment is no longer present in the revised version in the article. However, we have followed this advice to modify Fig. 7 (old Fig. 6) so that x-axis ticklabels appear on every panel.

**Fig. 7 caption - Please state what the white arrows represent in panel (c). They are defined on line 446 but should also be defined in the figure caption. Also, what is their magnitude? Also, add that these panels are for the Amazon box only.**

We thank the reviewer for this remark. However, after reading the major revisions we believe this figure is no longer pertinent for the manuscript.

**Line 472-473 - I think the authors mean to refer to Fig. 8c, the temperature tendency panel, and Fig. 8d, the horizontal temperature advection panel, in this sentence.**

We thank the reviewer for this comment. We have checked the figure references and have adapted them to the new numbering of the figures of the paper. Please, find the modifications between lines 518 and 521 of the new version.